# Spatial and Temporal Long-Term Patterns of Phyto and Zooplankton in the W-Mediterranean: RADMED Project

**María del Carmen García-Martínez [1],\*** , **Manuel Vargas-Yáñez [1]**, **Francina Moya [1]**,
**Rocío Santiago [2]**, **Andreas Reul [3]**, **María Muñoz [1]**, **José Luis López-Jurado [2]** and **Rosa Balbín [2]**

[1]  Instituto Español de Oceanografía, C.O. Málaga., Puerto Pesquero s/n, Fuengirola, 29640 Málaga, Spain;
     manolo.vargas@ieo.es (M.V.-Y.); francina.moya@ieo.es (F.M.); maria.munoz@ieo.es (M.M.)

[2]  Instituto Español de Oceanografía, C.O. Baleares, Muelle de Poniente s/n, 07015 Palma de Mallorca, Spain;
     rocio.santiago@ieo.es (R.S.); lopez.jurado@ieo.es (J.L.L.-J.); rosa.balbin@ieo.es (R.B.)

[3]  Andalucía Tech, Departamento de Ecología y Geología, Universidad de Málaga, Campus de Teatinos s/n,
     29071 Málaga, Spain; areul@uma.es

\*   Correspondence: mcarmen.garcia@ieo.es

**Abstract:** It is widely accepted that the Mediterranean is an oligotrophic sea where winter mixing favors the proliferation of diatoms and high values of zooplanktonic biomass, mainly associated with the growth of copepods. Stratified conditions from mid-spring to late autumn are dominated by the picophytoplanktonic groups and the increment of cladoceran abundances. This general picture has important exceptions. A regionalization of the Mediterranean Sea can be established, distinguishing oligotrophic and mesotrophic areas and different blooming periods. The RADMED monitoring program covers a large area from the southwestern limit of the Mediterranean to the Catalan Sea. The analysis of phyto and zooplankton time series extending from 1992 to 2016 in some cases, and from 2007 to 2016 in others, have shown that the Spanish Mediterranean waters have differentiated areas and trophic regimes as a result of the existence of several fertilizing mechanisms which include winter mixing, tidal mixing in the Strait of Gibraltar, cyclonic circulation cells and frontal systems. The present work describes these different mechanisms acting on the Spanish Mediterranean waters, and also the potentiality of monitoring programs for providing statistics suitable for operational activities or the initialization/validation of ecological models.

**Keywords:** phytoplankton; zooplankton; Western Mediterranean; spatial patterns; seasonal cycle; long-term changes

## 1. Introduction

The oceans play a key role in the current climate change scenario. From a physical point of view, they have stored more than 90% of the heat absorbed by the Earth since the mid twentieth century [1] and deep water formation processes are capable of transferring large amounts of $CO_2$ to the deep ocean because of the higher solubility of gases in cold winter waters [2]. From a biological point of view, wind-driven upwelling processes, mixing of the upper water column caused by winter stormy activity, and deep water convection inject nutrients to the photic layer increasing the primary production. A fraction of this new production eventually reaches the deep waters being transformed into $CO_2$ and nutrients by the action of nitrifying bacteria. In this way, the so called solubility and biological pumps contribute to the sequestration of $CO_2$ [3] absorbing about 30% of the human $CO_2$ emissions during past decades [4,5].

The warming of the upper layer of the ocean would increase its thermal stratification reducing the efficiency of the winter mixing and deep convection processes, decreasing the ventilation of the deep layers, the productivity of the sea and the sequestration of $CO_2$ [6–8]. Therefore, it is expected that the oxygen and nutrient distributions and the first steps of the food webs, that is, the phyto and zooplanktonic communities, will be affected by the warming of the oceans. Any change in the phyto and zooplankton communities would finally have an impact on the rest of the marine ecosystems through bottom-up trophic cascade effects [9].

The Mediterranean Sea is not an exception within the scenario depicted above [10,11]. On the contrary, it has been suggested that because of its reduced dimensions and its location, between three continents, with an increasing pressure from the touristic, agriculture and industrial sector, it could be especially vulnerable to climate change as well as to other anthropogenic stressors [12].

Detecting lasting changes in marine ecosystems requires the long-term monitoring of variables that could be considered as indicators of the environmental state of the sea. Then, average seasonal cycles can be calculated and long term changes for such cycles could be detected. Nevertheless, long-term monitoring programs in the Mediterranean Sea are scarce and data availability for physical, chemical and biological variables is uneven. Temperature and salinity data for the twentieth century are suitable for the detection of long-term trends in the intermediate and deep Mediterranean waters, but are not appropriate for the analysis of the upper layer where the high frequency variability is stronger than that of deeper levels and can mask such long-term changes [13–15]. The deployment of autonomous devices operating both in real time or delayed mode, such as profilers within the MedArgo program [16], or moored conductivity-temperature-depthmonitors (CTDs) in the Hydrochanges program [17], and the compilation of hydrographic data in several platforms such as Copernicus [18] or SeaDataNet [19] have helped to improve the availability of temperature and salinity data for the twenty-first century. Nevertheless, the study of most of the biochemical variables as well as the abundance and taxonomic composition of the phyto and zooplankton communities are based on in situ sampling by means of oceanographic campaigns.

Unfortunately, biological long-term time series in the Mediterranean Sea are scarce. In the particular case of the Western Mediterranean, very few biogeochemical monitoring programs do exist. To our knowledge, some of these few monitoring stations are the LTER (Long Term Ecological Research) Mare Chiara station, in the Bay of Naples [20–22], the Dyfamed station, in the Ligurian Sea [23,24], the Blanes Bay Microbial Observatory in the Blanes Bay (Catalan Sea coastal waters, [25]), the PHYTOCLY station in Calvi Bay (Corsica, [11,26]) and the zooplankton time series at the Station B at Villefranche [27,28]. The time series obtained at these sites have allowed us to improve our understanding of marine ecosystems. They have provided detailed descriptions of the seasonal cycles and trends of temperature and salinity, the dissolved oxygen and nutrient dynamics as well as the changes in the abundances and taxonomic composition of the phyto and zooplankton assemblages [20–22,25,27]. When dealing with more extended areas, our knowledge about such seasonal dynamics usually comes from the analysis of data gathered in research projects covering almost one seasonal cycle. Typically, the comparison of data from two or at most four oceanographic cruises, covering the different seasons, has allowed us to infer the main traits of biochemical and planktonic cycles in areas such as the Catalan and Balearic Seas [29–31] the Alboran Sea [32,33], or the whole Mediterranean [34].

The Instituto Español de Oceanografía (IEO, Spanish Institute for Oceanography) initiated multidisciplinary monitoring programs in 1992, 1994 and 1996 in the areas of Malaga Bay, at the northwestern sector of the Alboran Sea (ECOMALAGA project), to the south of Cape Palos (ECOMURCIA project), to the south of Mallorca Island (ECOBALEARES project) and in the Ibiza and Mallorca Channels (CIRBAL project, Circulación Regional en las aguas de las Islas BAleares). These programs have been unified and extended to the Catalan Sea in 2007 under the umbrella of the new monitoring program RADMED, Series Temporales de Datos Oceanográficos en el Mediterráneo (Figure 1 [35]). Previous works have shown the capability of RADMED for providing a description of

the seasonal dynamics of the physical properties of the water masses around the Spanish continental shelf and slope and to detect long term trends in the intermediate and deep waters of the Western Mediterranean [13,15]. Previous works have grouped the RADMED time series by seasons and estimated the mean values corresponding to each season for the concentrations of dissolved oxygen, nutrients and chlorophyll [36]. Besides this description of the average seasonal cycles, these authors analyzed the existence of decadal changes since the mid-1990s.

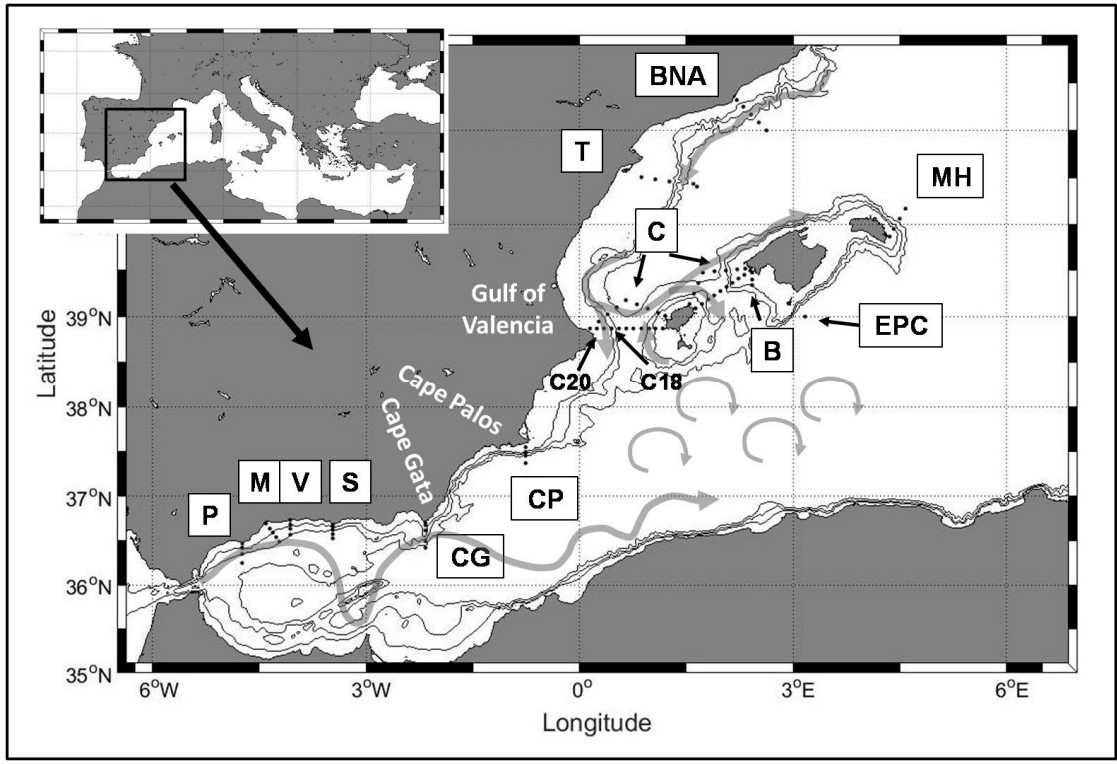

**Figure 1.** RADMED transects and stations. Grey lines are the basic features of the surface layer circulation (0–150 m).

The goal of the present work is to provide baseline data to support water quality and aquatic living resource characterization of the Spanish Mediterranean ecosystem. The datasets can further support development and validation of ecological and management-decision models of the ecosystem.For this purpose, phyto and zooplankton distributions were analyzed as follows:

(i) Seasonal cycles were analyzed for the micro-phytoplankton (>20 µm) abundances on the Spanish continental shelf from Malaga, in the westernmost sector of the Alboran Sea, to Barcelona (Catalan Sea), including the Balearic Islands.

(ii) The average cycles were estimated for the abundances of the nano (2–20 µm) and pico-eukaryote and pico-prokaryote (0.2–2 µm) phytoplankton on the continental shelf and slope.

(iii) Decadal changes in the abundances of micro-phytoplanktonic groups and in the meso-zooplanktonic biomass were studied at the stations where the time series extension made it possible.

(iv) The seasonal cycles of meso-zooplanktonic (>250 µm) biomass andabundances of broad meso-zooplanktonic groups were obtained.

In some cases, the time series analyzed will be more than 20 years long. In these cases the mean values which characterize the average seasonal cycles and the standard deviations and ranges of variability (minimum and maximum values recorded) can be considered as robust statistics. For the case of such long time series, decadal changes can also be estimated fitting a straight line by means

of least squares fit. At other stations, only a complete seasonal cycle (one data set per season) is available, and results should be considered with caution. In these latter cases the results are presented for the completeness of the work and are discussed in light of previous works also dealing with four cruises distributed along one single year. Section 2 is a detailed description of the sampling and analysis strategy within the RADMED project. The main results are in Section 3 and a discussion and conclusions are presented in Section 4.

## 2. Materials and Methods

The RADMED project is a monitoring program funded by the IEO covering the Spanish Mediterranean coast and Balearic waters (Figure 1). As mentioned above, this program was launched in 2007 unifying and extending previous monitoring programs initiated in 1992, 1994 and 1996. The sampling stations are distributed in transects perpendicular to the coast, covering the continental shelf and slope and in some cases including some deep stations (>2000 m). Stations are named by a letter corresponding to each transect and a number increasing from the coast to the open sea. In the Alboran Sea, the westernmost transects are Cape Pino (P in Figure 1), Málaga (M) and Vélez (V). For instance, the closest station to the coast in Cape Pino transect is named as P1, and the most offshore station is P4. Sacratif transect extends from Cape Sacratif in the central part of the Alboran Sea, and Cape Gata transect (CG) is on its eastern limit. Those transects extending from the eastern Spanish coast are Cape Palos (CP), Tarragona (T) and Barcelona (BNA). Two more transects are located in the Balearic Islands, one of them to the south of Mallorca Island (B) and another one extending in a northeast direction from Mahón, in Menorca Island (MH). Apart from these transects, there are 37 oceanographic stations forming two triangles covering the Balearic channels: the Ibiza Channel, between the Spanish Peninsula and Ibiza Island, and the Mallorca Channel, between Ibiza and Mallorca. These stations are labeled as C. Finally, a deep station (>2200 m) is located to the south of Cabrera Island (EPC). All the stations are visited every three months, that is, once per season, when vessel availability and weather conditions make it possible.

CTD (conductivity-temperature-depth) profiles were obtained in all the stations. The hydrographic sampling was done using CTDs, mainly the model SBE (Sea-Bird Scientific) 911 installed in a carousel water sampler SBE 32. Some profiles were obtained using a CTD of the model SBE 25. CTDs were equipped with a dissolved oxygen (DO) SBE 43 sensor. DO and conductivity sensors were calibrated using water samples from selected stations and depths. Water samples for the determination of nutrients and chlorophyll-*a* were taken at all the stations. Water samples for nutrient determinations were taken at 0, 10, 20, 50, 75, 100, 200, 300, 500, 700, 1000 m and sea bottom for deep stations while sampling was limited to the station depth for the shallower ones. Samples for the determination of chlorophyll concentrations were limited to the upper 100 m of the water column. As already commented, 37 oceanographic stations cover the Ibiza and Mallorca Channels. Such stations are distributed forming two triangles. Stations C20 and C18 within the Ibiza Channel triangle (see Figure 1), were chosen as representative of the peninsular continental shelf and slope conditions respectively (see Figure 1). Details concerning nutrients and chlorophyll sampling can be found in previous works [35,36].

Additionally to the standard sampling at each station (temperature, salinity, Dissolved Oxygen, chlorophyll-*a*, and nutrient concentrations), water samples for the determination of the micro-phytoplankton abundance were taken at the second station of each transect, e.g., P2, M2, V2, etc., located on the continental shelf. Samples for the determination of nano and picoeukaryote plankton abundance and prokaryote pico-phytoplankton abundance (*Prochlorococcus* and *Synechococcus* cyanobacteria) were taken at the second (continental shelf) and fourth (continental slope) stations of each transect (e.g., P2, P4, M2, M4, etc.). Water samples for micro-phytoplankton studies were obtained with 5L Niskin bottles from the sea surface to 100 m at the same discrete depths as chlorophyll and nutrient samples. Micro-phytoplankton samples were preserved in 150 mL bottles and fixed with 2 mL of acid Lugol solution and stored in darkness at ambient temperature until they were analyzed.

Cell counts were performed using an inverted microscope after sedimentation of variable volumes of seawater (25–100 mL), depending on cell concentration [37]. First, cells from subsamples were counted over the whole bottom of the sedimentation chamber scanned at $50\times$ magnification. Secondly, on two transects of the whole bottom area of the sedimentation chamber at $100\times$ magnification, and finally, on one transect of the whole bottom area of the sedimentation chamber at $200\times$ magnification. Determinations were done to genera level in most of the cases and to species level when possible. Nevertheless, considering the long-term objective of this project and that this task has a great time and personnel demand, time series were finally constructed for main groups. Diatoms, dinoflagellates, small flagellates and ciliates are well preserved within Lugol solutions. By contrast, coccolithophorids suffer a rapid dissolution. Good preservation of coccolithophorid cells has been obtained with very weak formaldehyde solutions buffered with hexamethylenetetramine [38,39]. This means having to store twice the volume of samples and double the examination work, which cannot be achieved due to the lack of space and qualified personnel during the campaigns and at the laboratory. For these reasons, the final time series in RADMED project were those of abundances of diatoms, dinoflagellates and small flagellates.

Pico and nano-phytoplankton samples were preserved in 5 mL cryotubes fixed with 200 µL of 50% glutaraldehyde solution and frozen at $-80\,^\circ$C until they were analyzed using a FACSCalibur (Becton Dickinson) flow cytometer. The pico-phytoplankton (both eukaryote and prokaryote) and nano-phytoeukaryote groups were described and enumerated according to their specific auto-fluorescence properties and side scatter differences [40].

Zooplankton samples were obtained at the second and fourth stations of each transect. Oblique hauls were carried out between the bottom (for stations <100 m depth) or 100 m depth (for stations >100 m depth) and the sea surface at a vessel speed of two knots with a Bongo-20 Plankton net fitted with 250 µm to sample meso-zooplankton. A general Oceanics flowmeter was fitted to the net to estimate the distance traveled during the oblique haul. Then the volume of filtered water was calculated using the net diameter and distance.

Immediately after collection, the zooplankton samples were split in two subsamples for biomass and taxonomic analysis with a Folsom plankton splitter. Subsamples for taxonomic studies were preserved with 4% neutralized formaldehyde buffered with hexamethylenetetramine. Representative aliquots for taxonomic analysis were analyzed and the organisms identified to the level of the main taxonomic groups: copepods, appendicularians, cladocerans, doliolids, chaetognaths, siphonofores, ostracods and scyphozoans. Abundance was calculated as individuals/m$^3$. Subsamples for biomass were frozen ($-20\,^\circ$C) for subsequent estimation of the biomass as dry weight [41,42]. Samples for biomass measurements were concentrated on Whatman GF/C filters. The mass of the filters had been previously recorded. These filters were then dried in a drying oven at $60\,^\circ$C for about 24 h and stored in a desiccator for approximately 1 h. Then biomass values were estimated by subtracting the initial mass of the filter from the final mass. Biomass was calculated as mg/m$^3$.

*Statistical Analyses*

The main objective of the RADMED project is to estimate basic statistics such as average seasonal values, ranges of variability and long-term trends of variablesthat could be used as indicators of possible natural and/or anthropogenic alterations, and help to a better understanding of the physical and biochemical dynamics of the Mediterranean waters. Such variables should satisfy two main requirements. The first one is that they should provide some insight about the environmental state of the Spanish Mediterranean waters, and second, that they should be easily monitored with a long-term perspective, maximizing data acquisition while optimizing the economic efficiency of the monitoring program. The present work analyzes all the time series collected for micro-phytoplankton, nano and picoeukaryote cell abundances and prokaryote pico-phytoplankton cells (*Synechococcus* and *Prochlorococcus* cyanobacteria) from the sea surface to 100 m depth. For each depth, mean values were estimated for each season. Standard deviations and the minimum and maximum values recorded

were also obtained for characterizing the time variability associated to each variable and season. Once the seasonal cycle was estimated, it was subtracted to the original data and time series of residuals or deviation were obtained. Linear trends were estimated by least squares fits and confidence intervals at the 5% significance level (95% confidence level) were calculated using a t-student distribution for the slope after checking for the normality of the residuals. The same analyses, mean values, standard deviation and range of variability were accomplished for the abundances of the main meso-zooplanktonic groups and biomass. In this case it was not possible to determine the vertical distribution of abundances and biomasses as the data proceeded from oblique hauls from 100 m to the sea surface.

A monitoring program such as RADMED faces different difficulties such as weather conditions which make the sampling impossible in some cases, instrument failures, or simply the lack of vessel availability. In other cases there are difficulties to have the needed personnel for the analysis of a huge number of samples. Besides this, the RADMED program was initiated in 2007, unifying some previous more local monitoring projects. Therefore, data previous to, and after 2007 have frequent gaps and different temporal coverage depending on the station considered. Table 1 shows this coverage for each variable and station. Graphical representations of the seasonal cycles for phytoplankton, zooplankton and prokaryote pico-phytoplankton are presented when at least one seasonal cycle is available. Nevertheless, detailed tables presenting robust statistics that could be considered as the most likely seasonal dynamics will be presented only when the data availability allow such calculations. Linear trends for the detection of possible long-term changes will only be presented for those stations initiated in the 1990s.

## 3. Results

### 3.1. Micro-Phytoplankton

#### 3.1.1. Spatial and Seasonal Patterns

The abundances of micro-phytoplankton showed important differences from southwest (Alboran Sea) to northeast (Catalan Sea). Figures 2–5 show the vertical distributions of micro-phytoplankton, pico and nanoplankton abundances at four transects representing such differences (transects P, C, B and BNA). Station P2, which is the closest one to the Strait of Gibraltar showed diatom abundances higher than the other two groups (dinoflagellates and small flagellates) from spring to autumn, with abundances that could reach 300 and 400 cel./mL at the upper 20 m of the water column. Only in winter small flagellates were the most abundant group along the water column, with the only exception of 50 m depth where diatom concentrations were higher (Figure 2A). The highest abundances for the three groups were observed at the sea surface or at the upper part of the water column (10 or 20 m depth), and then decreased with depth, the only exception being the deep diatom maximum at 50 m depth in winter. Transects M and V (not shown) located around Malaga Bay, presented a similar behavior to the one observed in transect P. Diatom abundances decreased eastwards at the same time that small flagellates increased. Diatoms were the most abundant group in stations S2 and CG2 during winter, while small flagellates dominated from spring to autumn (Figure 6). Maximum values were observed at the sea surface or at the upper 20 m of the water column for both diatoms and small flagellates and for the whole Alboran Sea. Dinoflagellate abundances were the lowest ones and their vertical distribution seemed to be more homogeneous (see Figure 2C,E,G).

**Table 1.** Data available for each variable (M: micro-phytoplankton, C: pico and nanoplankton by citometry; Z.A. zooplankton abundance and Z.B. zooplankton biomass) (rows) at each transect, and year (columns). Grey color indicates data available. Every year is divided in four columns, one per season (winter, spring, summer and autumn). On the left the name/s of the transect: P (Cape Pino), M (Malaga), V(Vélez), S (Sacratif), CG (Cape Gata), CP (Cape Palos), C (Ibiza Channel, peninsular side), T(Tarragona), BNA(Barcelona), B (Mallorca transect) and MH (Mahon).

| Transect | Variable | 1992 | 1993 | 1994 | 1995 | 1996 | 1997 | 1998 | 1999 | 2000 | 2001 | 2002 | 2003 |
|---|---|---|---|---|---|---|---|---|---|---|---|---|---|
| **P,M,V** | M | | | | | | | | | | | | |
| | C | | | | | | | | | | | | |
| | Z.A. | | | | | | | | | | | | |
| | Z.B. | | | | | | | | | | | | |

| Transect | Variable | 2004 | 2005 | 2006 | 2007 | 2008 | 2009 | 2010 | 2011 | 2012 | 2013 | 2014 | 2015 | 2016 |
|---|---|---|---|---|---|---|---|---|---|---|---|---|---|---|
| **P,M,V** | M | | | | | | | | | | | | | |
| | C | | | | | | | | | | | | | |
| | Z.A. | | | | | | | | | | | | | |
| | Z.B. | | | | | | | | | | | | | |

| Transect | Variable | 2004 | 2005 | 2006 | 2007 | 2008 | 2009 | 2010 | 2011 | 2012 | 2013 | 2014 | 2015 | 2016 |
|---|---|---|---|---|---|---|---|---|---|---|---|---|---|---|
| **S,CG** | M | | | | | | | | | | | | | |
| | C | | | | | | | | | | | | | |
| | Z.A. | | | | | | | | | | | | | |
| | Z.B. | | | | | | | | | | | | | |

| Transect | Variable | 2003 | 2004 | 2005 | 2006 | 2007 | 2008 | 2009 | 2010 | 2011 | 2012 | 2013 | 2014 | 2015 | 2016 |
|---|---|---|---|---|---|---|---|---|---|---|---|---|---|---|---|
| **CP** | M | | | | | | | | | | | | | | |
| | C | | | | | | | | | | | | | | |
| | Z.A. | | | | | | | | | | | | | | |
| | Z.B. | | | | | | | | | | | | | | |

| Transect | Variable | 2007 | 2008 | 2009 | 2010 | 2011 | 2012 | 2013 | 2014 | 2015 | 2016 |
|---|---|---|---|---|---|---|---|---|---|---|---|
| **C** | M | | | | | | | | | | |
| | CIT | | | | | | | | | | |
| | Z.A. | | | | | | | | | | |
| | Z.B. | | | | | | | | | | |

**Table 1.** *Cont.*

| T | | 2007 | 2008 | 2009 | 2010 | 2011 | 2012 | 2013 | 2014 | 2015 | 2016 |
|---|---|---|---|---|---|---|---|---|---|---|---|
| | M | | | | | | | | | | |
| | CIT | | | | | | | | | | |
| | Z.A. | | | | | | | | | | |
| | Z.B. | | | | | | | | | | |

| BNA | | 2007 | 2008 | 2009 | 2010 | 2011 | 2012 | 2013 | 2014 | 2015 | 2016 |
|---|---|---|---|---|---|---|---|---|---|---|---|
| | M | | | | | | | | | | |
| | CIT | | | | | | | | | | |
| | Z.A. | | | | | | | | | | |
| | Z.B. | | | | | | | | | | |

| B | | 1994 | 2007 | 2008 | 2009 | 2010 | 2011 | 2012 | 2013 | 2014 | 2015 | 2016 |
|---|---|---|---|---|---|---|---|---|---|---|---|---|
| | M | | | | | | | | | | | |
| | CIT | | | | | | | | | | | |
| | Z.A. | | | | | | | | | | | |
| | Z.B. | | | | | | | | | | | |

| MH | | 2007 | 2008 | 2009 | 2010 | 2011 | 2012 | 2013 | 2014 | 2015 | 2016 |
|---|---|---|---|---|---|---|---|---|---|---|---|
| | M | | | | | | | | | | |
| | CIT | | | | | | | | | | |
| | Z.A. | | | | | | | | | | |
| | Z.B. | | | | | | | | | | |

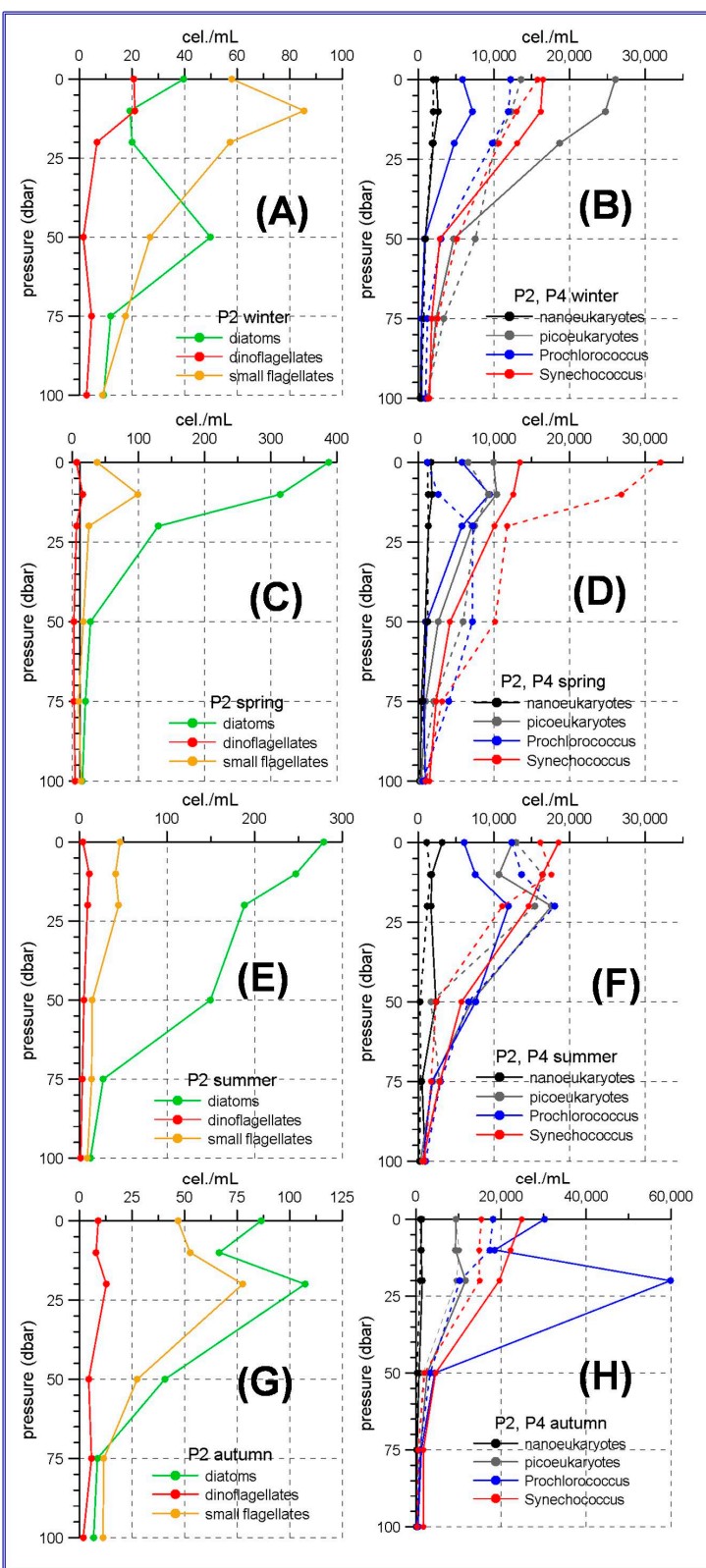

**Figure 2.** (**A**) shows the winter average vertical distribution of diatom abundances (green line), small flagellates (yellow line) and dinoflagellates (red line) at station P2. (**B**) shows the winter average vertical distribution for nanoeukaryotes (black line), picoeukaryotes (grey line), *Prochlorococcus* (blue line) and *Synechococcus* (red line) at the stations P2 (solid line) and P4 (dashed line). (**C**,**D**) are the same for spring, (**E**,**F**) for summer and (**G**,**H**) for autumn.

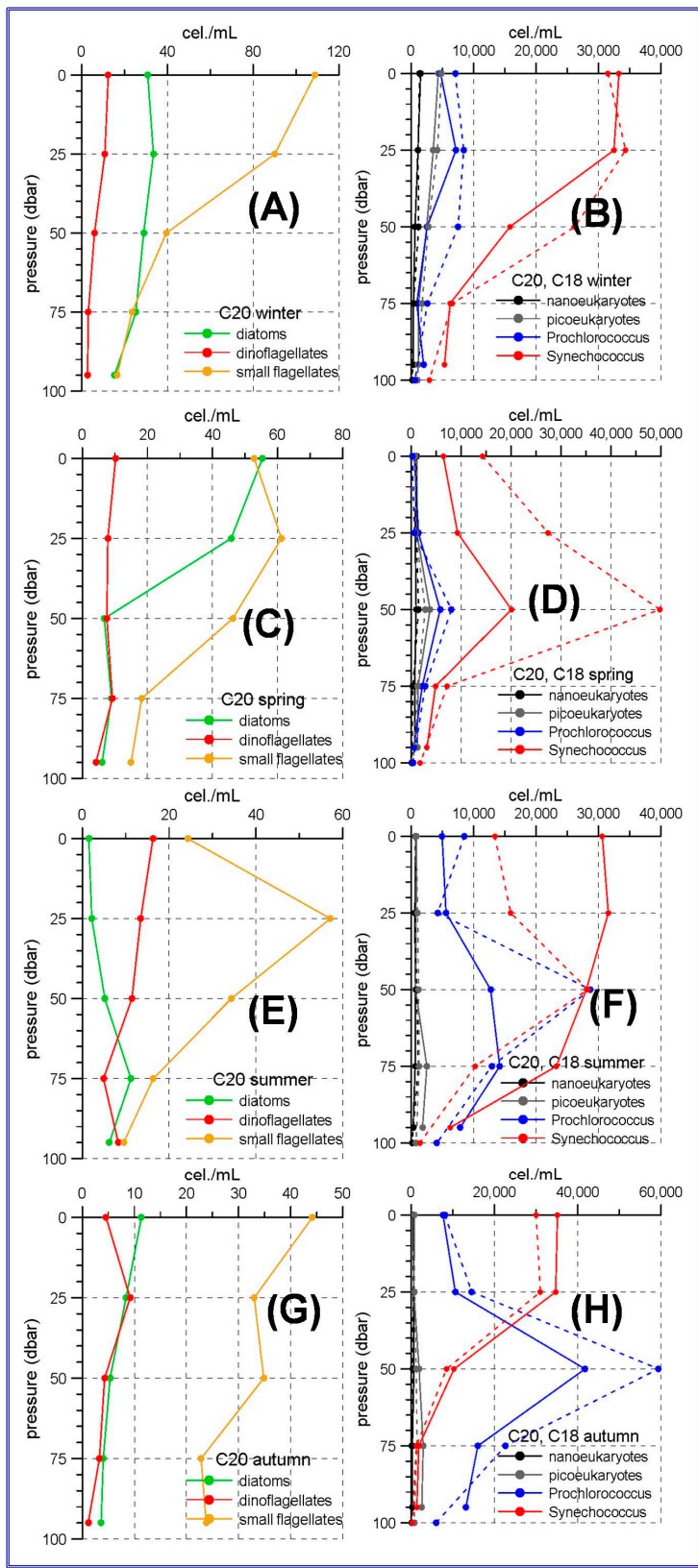

**Figure 3.** (**A**) shows the winter average vertical distribution of diatom abundances (green line), small flagellates (yellow line) and dinoflagellates (red line) at station C20. (**B**) shows the winter average vertical distribution for nanoeukaryotes (black line), picoeukaryotes (grey line), *Prochlorococcus* (blue line) and *Synechococcus* (red line) at the stations C20 (solid line) and C18 (dashed line). (**C**,**D**) are the same for spring, (**E**,**F**) for summer and (**G**,**H**) for autumn.

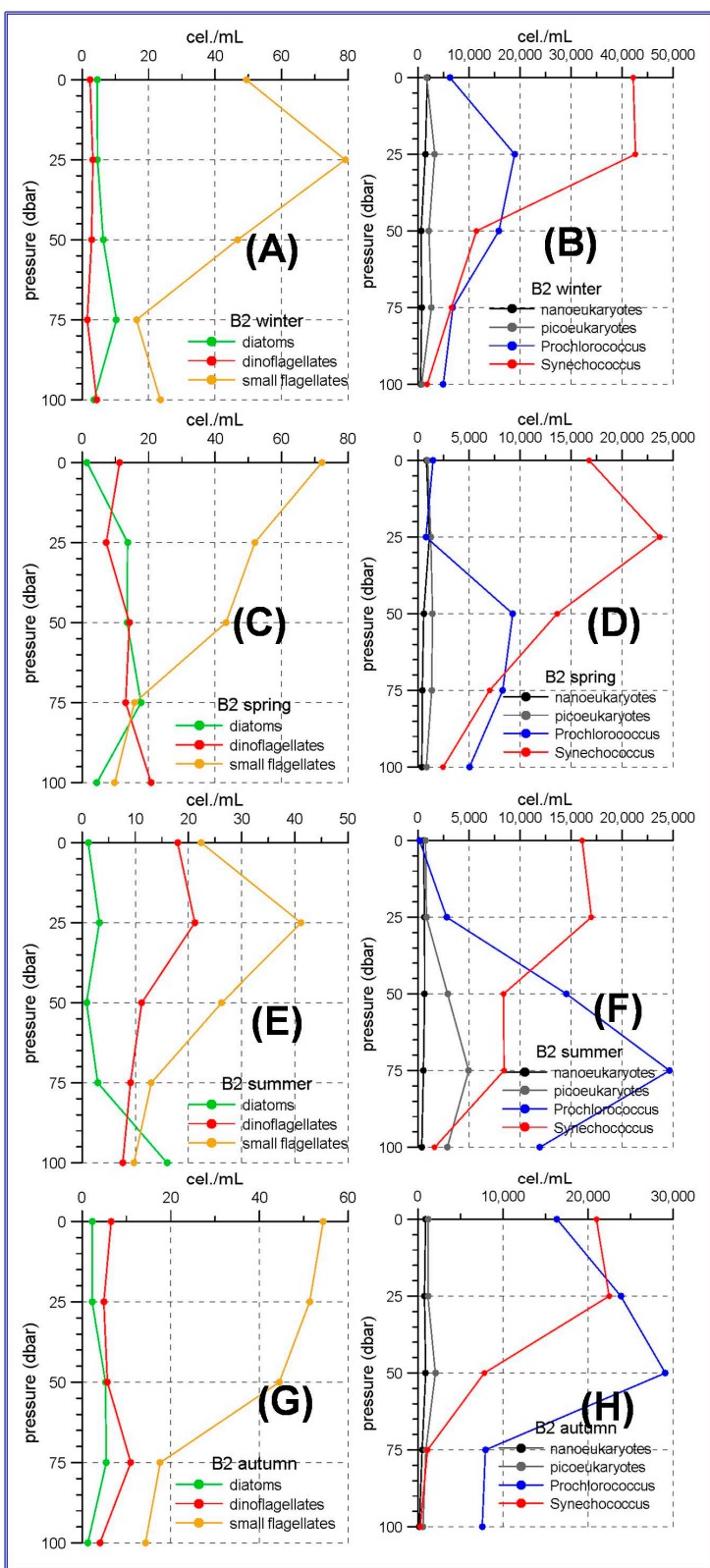

**Figure 4.** (**A**) shows the winter average vertical distribution of diatom abundances (green line), small flagellates (yellow line) and dinoflagellates (red line) at station B2. (**B**) shows the winter average vertical distribution for nanoeukaryotes (black line), picoeukaryotes (grey line), *Prochlorococcus* (blue line) and *Synechococcus* (red line) at the station B2 (solid line). (**C**,**D**)are the same for spring, (**E**,**F**) for summer and (**G**,**H**) for autumn.

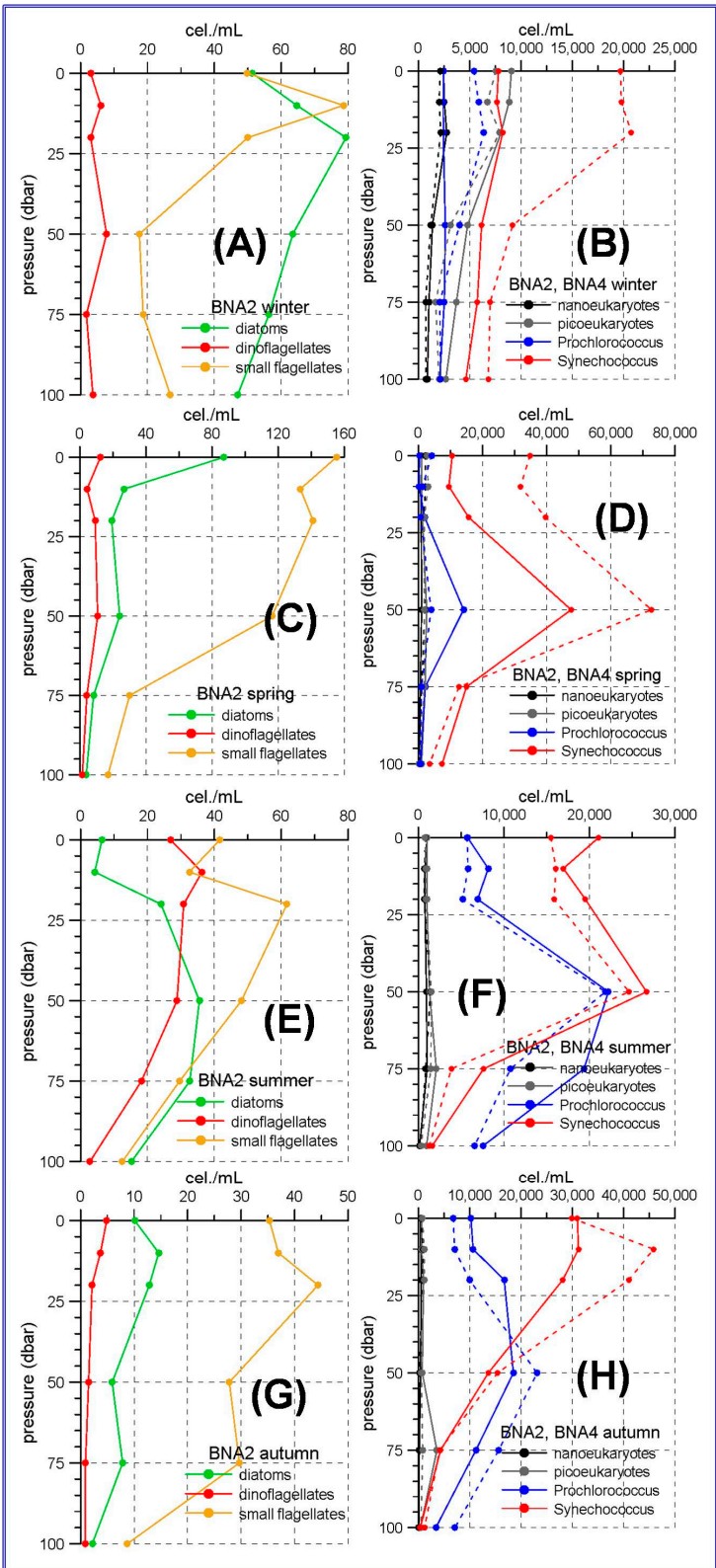

**Figure 5.** (**A**) shows the winter average vertical distribution of diatom abundances (green line), small flagellates (yellow line) and dinoflagellates (red line) at station BNA2. (**B**) shows the winter average vertical distribution for nanoeukaryotes (black line), picoeukaryotes (grey line), *Prochlorococcus* (blue line) and *Synechococcus* (red line) at the BNA2 station (solid line) and BNA4 station (dashed line). (**C,D**) are the same for spring, (**E,F**) for summer and (**G,H**) for autumn.

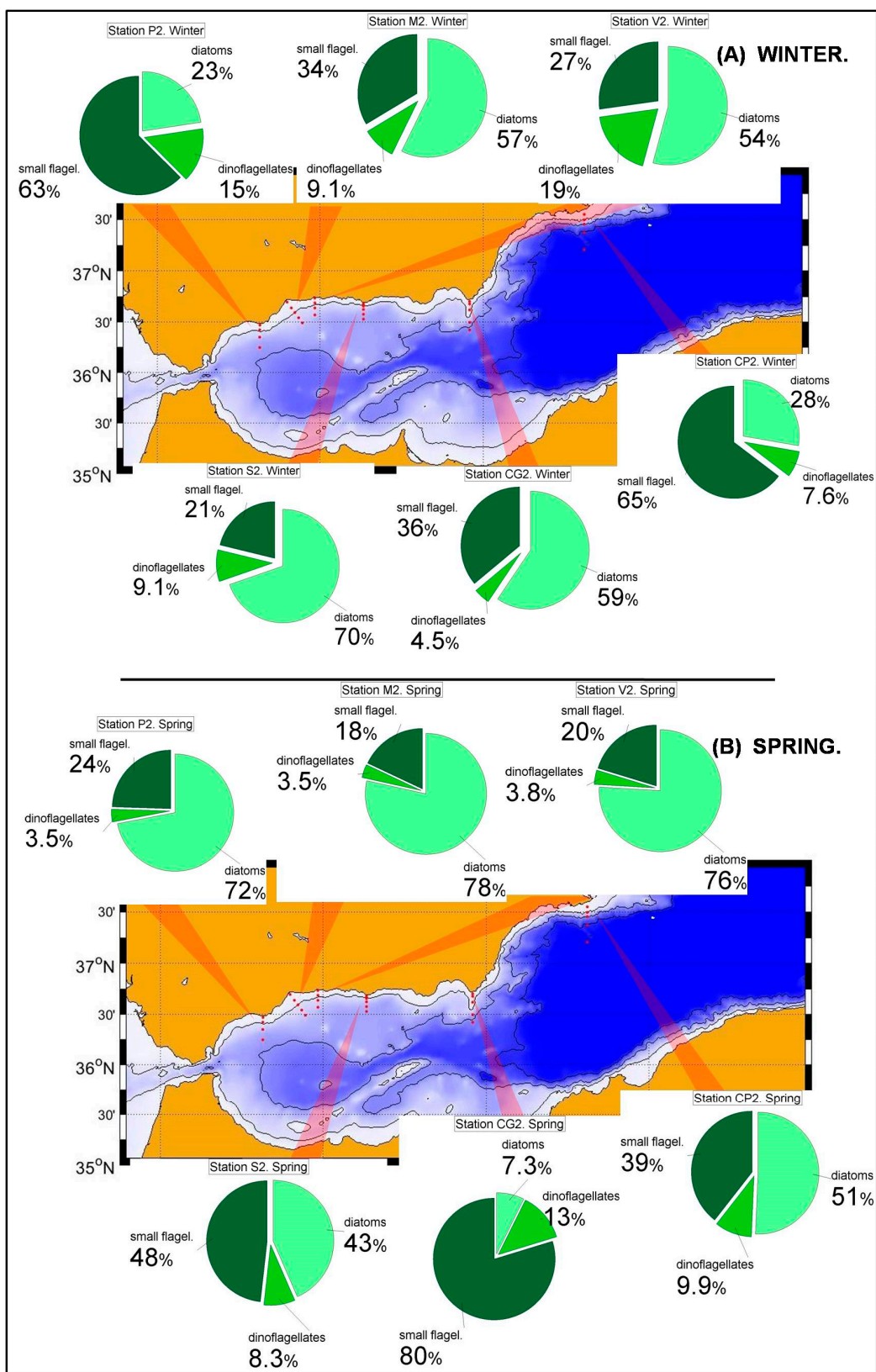

**Figure 6.** *Cont.*

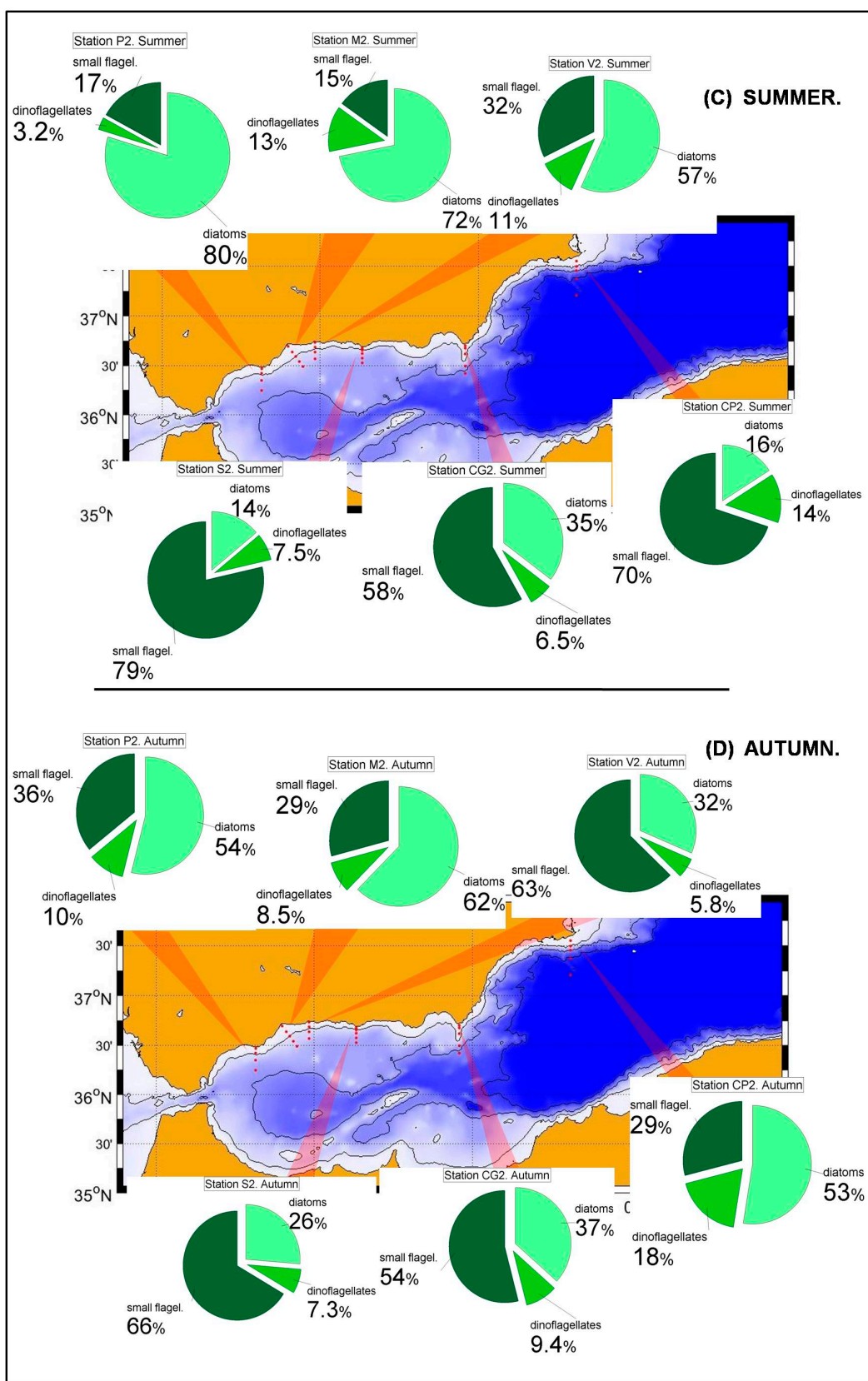

**Figure 6.** Relative abundances expressed as percentages for the Alboran Sea and Cape Palos for diatoms (light green), small flagellates (dark green) and dinoflagellates (intermediate green) integrated from the sea surface to 100 m. (**A–D**) correspond to winter, spring, summer and autumn respectively.

The eastward gradient, from transect P to transect CG, observed in the Alboran Sea stations, is confirmed by those transects to the northeast of Cape Gata. Figure 3 shows that the small flagellate group is the dominant one throughout the whole year in the peninsular side of the Ibiza Channel (station C20). The diatom abundances decreased considerably from those found in the Alboran Sea while the values for small flagellates and dinoflagellates were similar (compare Figures 2 and 3). In the waters surrounding the Balearic Islands, diatom abundances rarely exceeded 10 cel./mL throughout the year and for the whole water column, with small flagellates the main group (Figure 4A,C,E,G). The diatom maximum, although very weak, was located between 50 and 75 m (Figure 4C, G) or at 100 m depth (Figure 4E). The abundances of dinoflagellates and small flagellates seemed to be similar to those observed in the Alboran Sea, with the small flagellate maxima at the sea surface or at the upper 25 m of the water column. Dinoflagellates increased during summer (Figure 4E) when the abundances of this group were higher than those of the diatoms.

The station BNA2, in Barcelona transect, exhibited some common features to those observed in the Ibiza Channel and the Balearic Islands, confirming the southwest–northeast gradient already described (from the Alboran Sea to the north of Cape Gata), but also included some new features. From spring to autumn the small flagellate group was the most abundant, with abundances similar to those described for the rest of the RADMED area. Nevertheless, in winter the diatoms were the most abundant group along the water column with high abundances at the sea surface and a maximum at 20 m depth (Figure 5A). Diatoms developed a deep maximum between 50 and 75 m in summer. Small flagellate maxima were located at the surface or at 20 m depth, as observed in other transects. As in the case of station B2, dinoflagellates increased during summer (Figure 5E) with higher abundances than those for diatoms in the upper 20 m of the water column.

The latitudinal differences in the group distributions were more clearly observed using relative abundances of groups, integrated for the upper 100 m of the water column. Figure 6 shows that diatoms were the most abundant group throughout the whole year for the station M2, and from winter to summer for the station V2. Autumn diatom abundances for the station V2 were lower than those of the small flagellate ones, but still presented high values. Diatoms were also the dominant group from spring to autumn in station P2. The relative importance of diatoms decreased eastwards within the Alboran Sea. Diatoms were the most abundant group at the stations S2 and CG2 only in winter and in CP2 in spring and autumn. In all the cases, dinoflagellates were the least abundant group. Figure 7 shows the integrated relative abundances for the main micro-phytoplanktonic groups from the Ibiza Channel to the north. Diatoms were the most abundant group only in winter in the station BNA2, at the Catalan Sea continental shelf, and in summer in MH2 to the northeast of Menorca Island. Apart from these two cases, small flagellates were the most abundant group at these stations for the whole year. It is also interesting to note the increase of the dinoflagellate group in all the stations from C20 to the north, for the summer season, when this group became the second in importance behind the small flagellate one at some of the stations (Figure 7C).

Besides the main traits of the micro-phytoplankton seasonal cycles depicted above, one of the main goals of the RADMED project is to provide statistics that could be used as a reference for future works, numerical modeling validation and/or initialization, or any other operational services. Tables S1–S4 in the Supplementary Materials show the mean values, standard deviations and ranges of variability for the stations P2, M2, V2 and B1,which are some of the best sampled ones, with more than 10 seasonal values used for calculations in most of the cases.

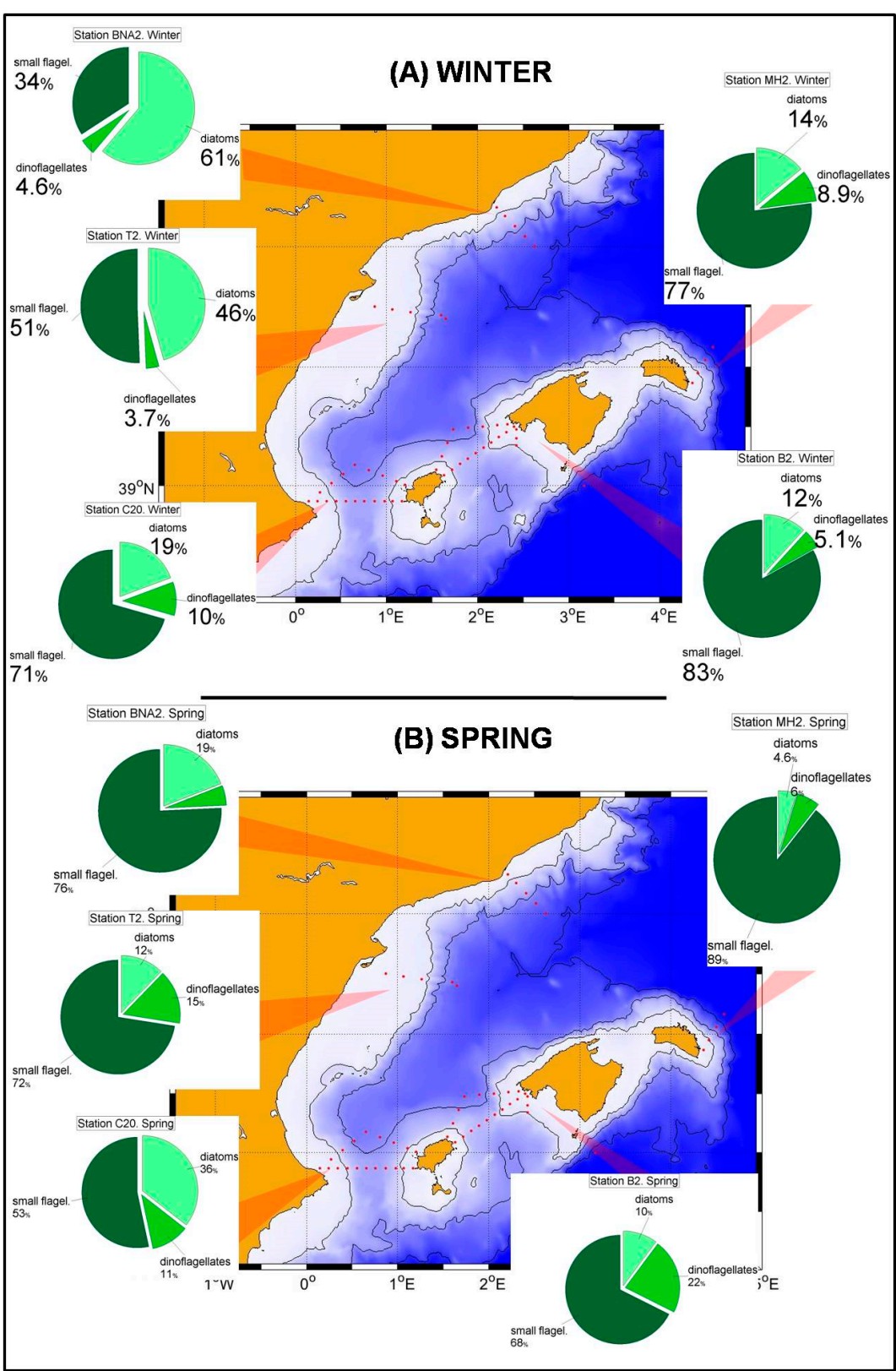

**Figure 7.** *Cont.*

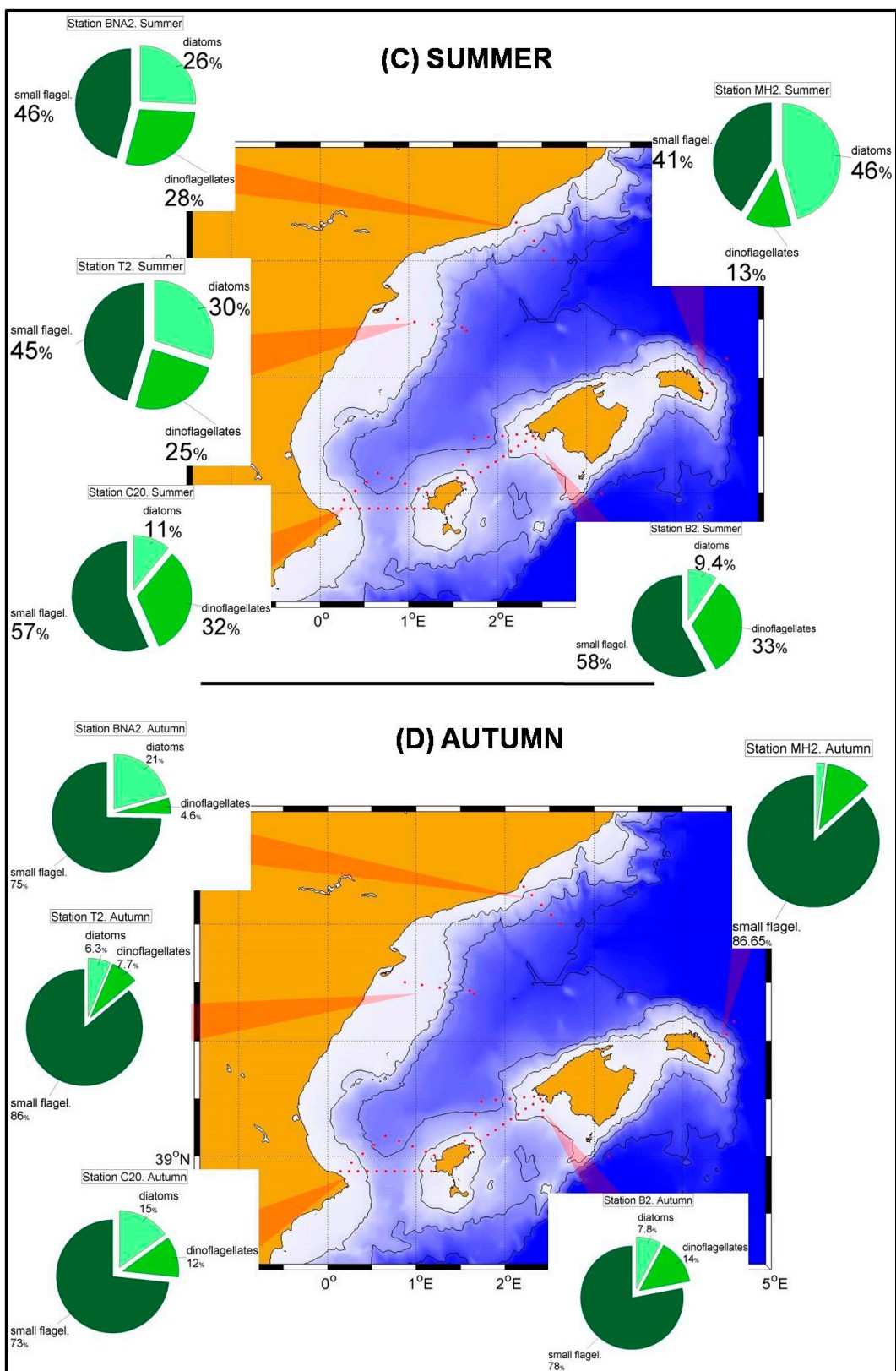

**Figure 7.** Relative abundances expressed as percentages for the transects to the north of Cape Palos for diatoms (light green), small flagellates (dark green) and dinoflagellates (intermediate green) integrated from the sea surface to 100 m. (**A**–**D**) correspond to winter, spring, summer and autumn respectively.

### 3.1.2. Micro-Phytoplankton Linear Trends

Micro-phytoplankton abundance time series were initiated in 1992 in the case of the westernmost Alboran Sea stations (P2, M2 and V2) and in 1994 in the case of station B1(Mallorca transect). Gaps and the elimination of outliers provided final time series with slightly different lengths to the nominal ones. Besides this, these time series cover more than 20 years and, therefore, an attempt to estimate long-term changes was addressed. Figures 8 and 9, show the residuals or deviations from the average seasonal cycle for the abundances of diatoms (Figure 8A,D and Figure 9A,D), dinoflagellates (Figure 8B,E and Figure 9B,E) and small flagellates (Figure 8C,F and Figure 9C,F). Time series of residuals are presented for 0 and 50 m depth. Nevertheless, the estimation of linear trends was accomplished for all the sampled depths (0, 10, 20, 50, 75 and 100 m, see Table S5 in Supplementary Materials).

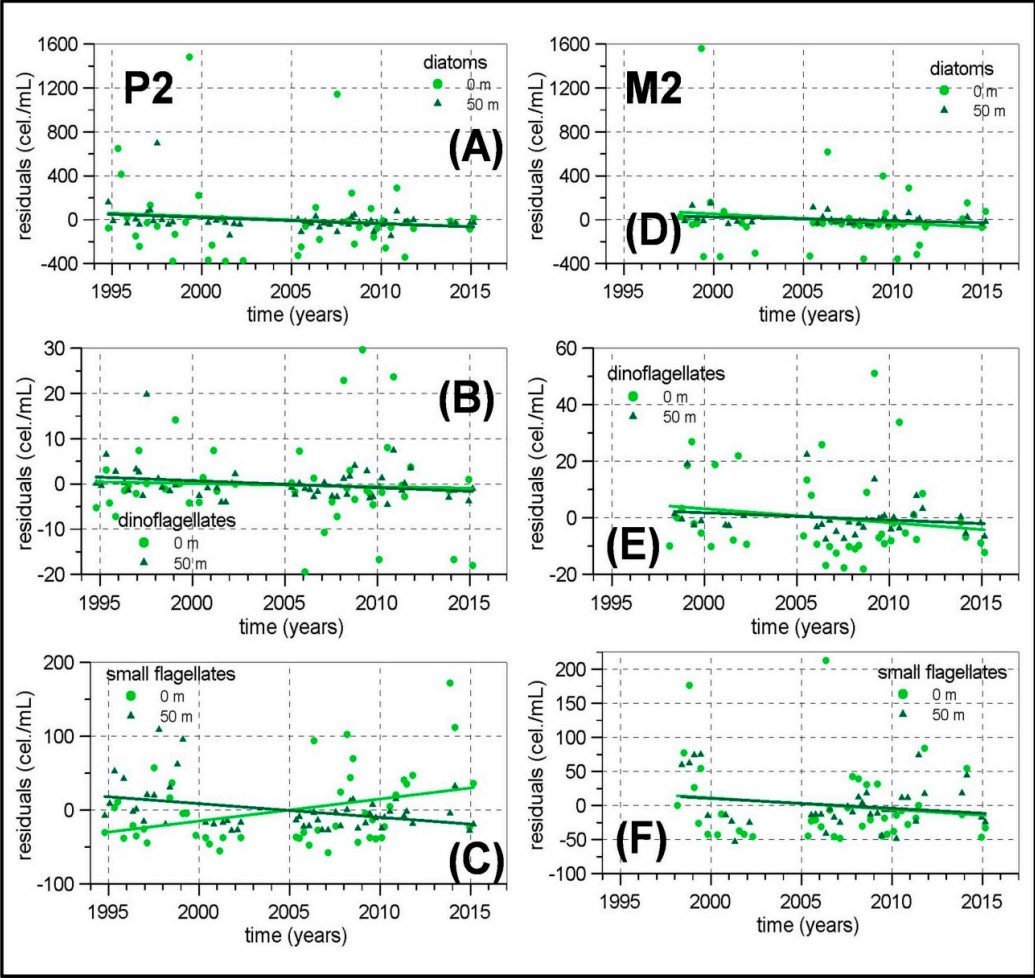

**Figure 8.** Time series of residuals of the micro-phytoplanktonic time series at stations P2 and M2. (**A–C**) correspond to diatoms, dinoflagellates and small flagellates at station P2. (**D–F**) correspond to diatoms, dinoflagellates and small flagellates at station M2. In all the figures light green dots are used for the time series of residuals at 0 m depth, and the dark green dots are used for the time series of residuals at 50 m depth. Solid lines are the straight lines fitted by means of least squares fit to each time series.

Diatoms in station P2 showed no significant trends at the 95% confidence level at 0, 10 and 20 m and a negative trend of $-5.7 \pm 5.5$ cel. mL$^{-1}$/year for the 50 m depth level. These time series extended from 1994 to 2016. A decreasing significant trend of $-1.6 \pm 1.3$ cel. mL$^{-1}$/year was obtained at 100 m depth, but in this case the series extended from 2005 to 2016. No significant trends were obtained for dinoflagellates. Small flagellates increased in a significant way at 0 m, showed a non-significant

trend at 10 m, and decreased at a rate of $-3.5 \pm 3.4$ and $-1.9 \pm 1.3$ cel. mL$^{-1}$/year for the 20 and 50 m levels. A positive trend was estimated at 75 m, but once again in this case the time series had a reduced length from 2005 to 2016. Diatom and dinoflagellate abundances in station M2 extended from 1997 to 2016 for all the depths from 0 to 75m. Diatom abundances decreased with trends of $-3.5 \pm 3.3$ cel. mL$^{-1}$/year and $-1.8 \pm 1.5$ cel. mL$^{-1}$/year at 50 and 75 m respectively. Dinoflagellate abundances showed no significant trends. Small flagellates decreased at the 75 m level with a trend of $-3.5 \pm 2.2$ cel. mL$^{-1}$/year for the period 1997–2016. Abundance time series in station V2 extended from 1997 to 2016. Diatom and dinoflagellate abundances did not change in a significant way. Small flagellates decreased at a rate of $-5.3 \pm 3.9$ cel. mL$^{-1}$/year at 75 m depth.

Time series at station B1 (Mallorca transect) extend from 1994 to 2015. Both diatom and small flagellate abundances increased along the whole water column with significant trends ranging from $0.3 \pm 0.2$ cel. mL$^{-1}$/year to $4.2 \pm 1.4$ cel. mL$^{-1}$/year. No significant changes were observed for the dinoflagellate abundances.

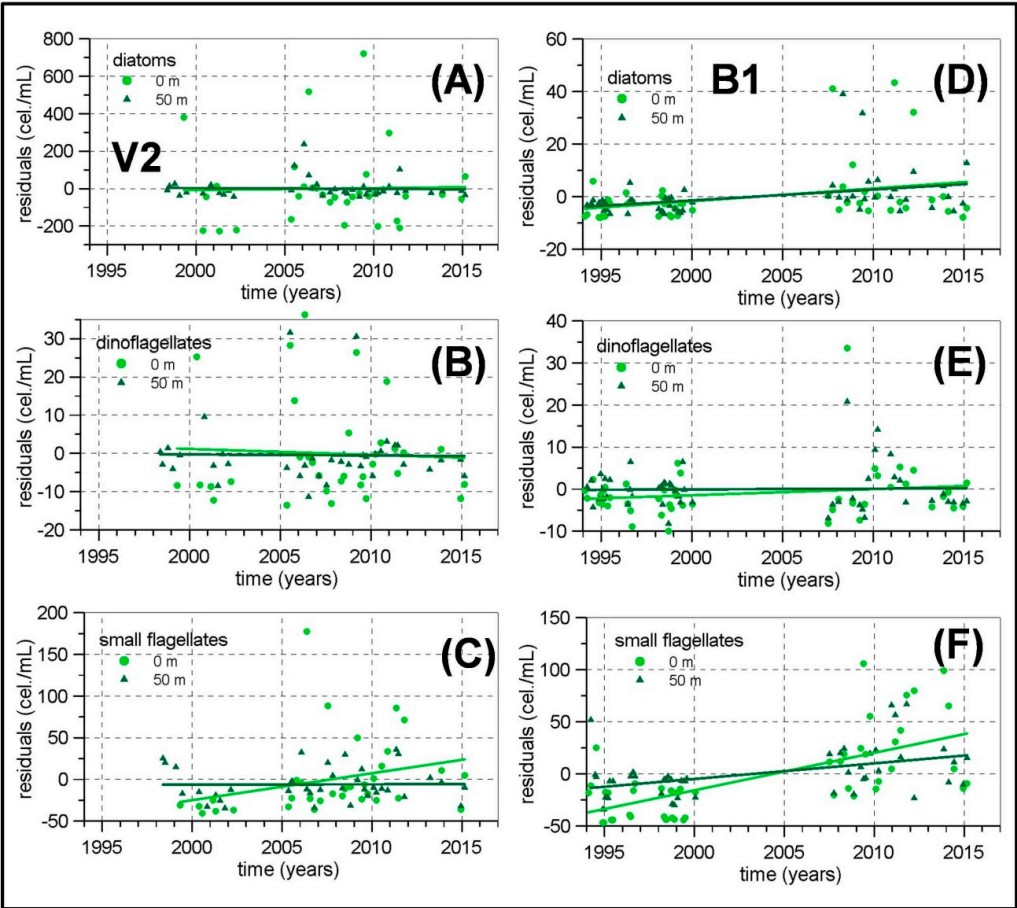

**Figure 9.** Time series of residuals of the micro-phytoplanktonic time series at stations V2 and B1. (**A**–**C**) correspond to diatoms, dinoflagellates and small flagellates at station V2. (**D**–**F**) correspond to diatoms, dinoflagellates and small flagellates at station B1. In all the figures light green dots are used for the time series of residuals at 0 m depth, and the dark green dots are used for the time series of residuals at 50 m depth. Solid lines are the straight lines fitted by means of least squares fit to each time series.

### 3.2. Nanoeukaryote, Picoeukaryote, Prochlorococcus and Synechococcus Abundances

The most abundant groups were *Synechococcus* and *Prochlorococcus* (prokaryote pico-phytoplankton), followed by picoeukaryotes (Figure 2B,D,F,H, Figure 3B,D,F,H, Figure 4B,D,F,H and Figure 5B,D,F,H). Prokaryotes showed abundances of the order of $10^4$ cel./mL. Nanoeukaryotes

were the less abundant group. Nano and picoeukaryote abundances were of the order of $10^3$ cel./mL in most of the cases, being stations in the Western Alboran Sea the only exceptions. For the latter stations picoeukaryote abundances reached similar values to those of the prokaryote pico-phytoplanktonic groups (Figure 2B,D,F,H, Figure 3B,D,F,H, Figure 4B,D,F,H and Figure 5B,D,F,H). Once again there are significant differences between the Alboran Sea and the waters to the north of Cape Gata. Despite the sharp *Prochlorococcus* maximum during autumn at 20 m depth in P2 (Figure 2H), or the weaker maximum in summer at the same depth (Figure 2F), the abundances of all these groups seemed to reach maximum values at the sea surface, decreasing with depth at stations P2 and P4. Some subsurface maxima were also found in the upper part of the water column (10, 20 m, Figure 2D,F). At the Ibiza Channel transect (Figure 3), *Synechococcus* reached the highest abundances at the sea surface during winter and autumn, showing a maximum at 25 m very close to the surface values during summer and at 50 m in spring. The vertical distributions were similar when stations C20 and C18 were compared and it cannot be established which of both stations (C20 and C18) showed higher abundances. *Prochlorococcus* always presented deep maxima between 25 and 50 m in winter, spring and autumn, and between 50 and 75 m for the summer season at C20. In all the cases the nano and picoeukaryote abundances were much lower and vertically uniform than the *Synechococcus* and *Prochlorococcus* ones. Station B2 (Figure 4) showed more clearly the traits commented above. During the whole year the *Synechococcus* abundances decreased from the upper 25 m of the water column to the sea bottom, while *Prochlorococcus* showed a deep maximum which deepened from 25 m in winter to 75 m in summer. Nano and picoeukaryotes had abundances one order of magnitude lower and their distributions were vertically more homogeneous. Finally, the station BNA2 and BNA4 (Figure 5) showed slight variations as in the case of micro-phytoplankton analyses. The general pattern for all the groups from spring to autumn (Figure 5D,F,H) was similar to that described for the stations of the Ibiza Channel and the Mallorca transect (C20, C18 and B2). Nevertheless, the winter values were somehow different, with more homogeneous vertical distributions decreasing from the surface to the bottom and pico and nano eukaryote abundances comparable to those of the prokaryote groups.

Tables S6–S8 in Supplementary Materials show the abundances of the eukaryote nano and pico-phytoplankton and the abundances of the prokaryote pico-phytoplankton for the stations P2, B2 and BNA2, representative of the three main areas described above.

### 3.3. Zooplankton Abundances and Biomass

Figure 10 shows the winter, spring, summer and autumn relative abundances of broad taxonomic groups in the Alboran Sea and in Cape Palos transect. In winter and spring copepods were the most abundant group with abundances higher than (P2, M2, V2, CG2 and CP2), or very close to (S2), the sum of all the other groups. In winter the second most abundant group in P2, M2, V2 and CG2 was appendicularians. Although station S2 showed the highest relative contribution of appendicularians (18.3%) among all the stations, the second most abundant group at this station was cladocerans (23.8%). The remaining groups (doliolids, chaetognaths, ostracods, siphonofores and scyphozoans) were less abundant and with fractions lower than 5% in most of the cases. In spring (Figure 10B) there is an increase of the relative contribution of cladocerans which became the second group at stations M2 and V2, although copepods continued to represent more than 50% of the total meso-zooplanktonic abundance (the only exception is station S2 where copepods represented just the 50% of the total abundance). The increase of the cladoceran abundance continued in summer (Figure 10C) when this group showed abundances similar or even higher (P2 and S2) than those of copepods. Doliolids became the third more abundant group in most of the stations. Notice that the chaetognath abundance increased eastwards (CG2 and CP2). In autumn (Figure 10D), although the cladoceran abundance was still high, being the second most abundant group in P2, M2, V2 and S2, copepod relative abundance increased and represented more than or close to the 50% of the total meso-zooplanktonic community at the cited stations, being the most abundant group at the rest of the stations. As in summer, chaetognath importance was higher towards the eastern part of the Alboran Sea and Cape Palos area.

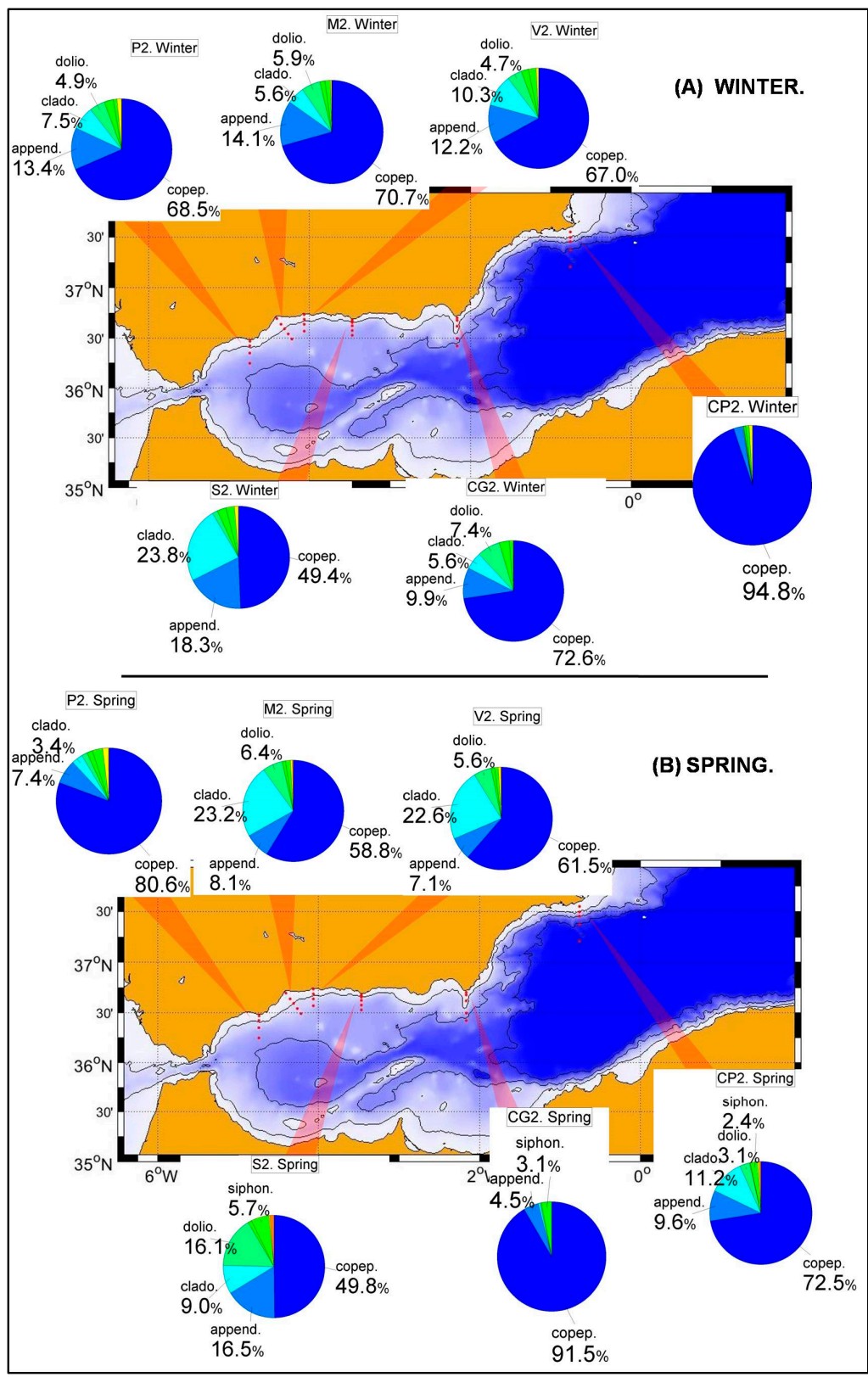

**Figure 10.** *Cont.*

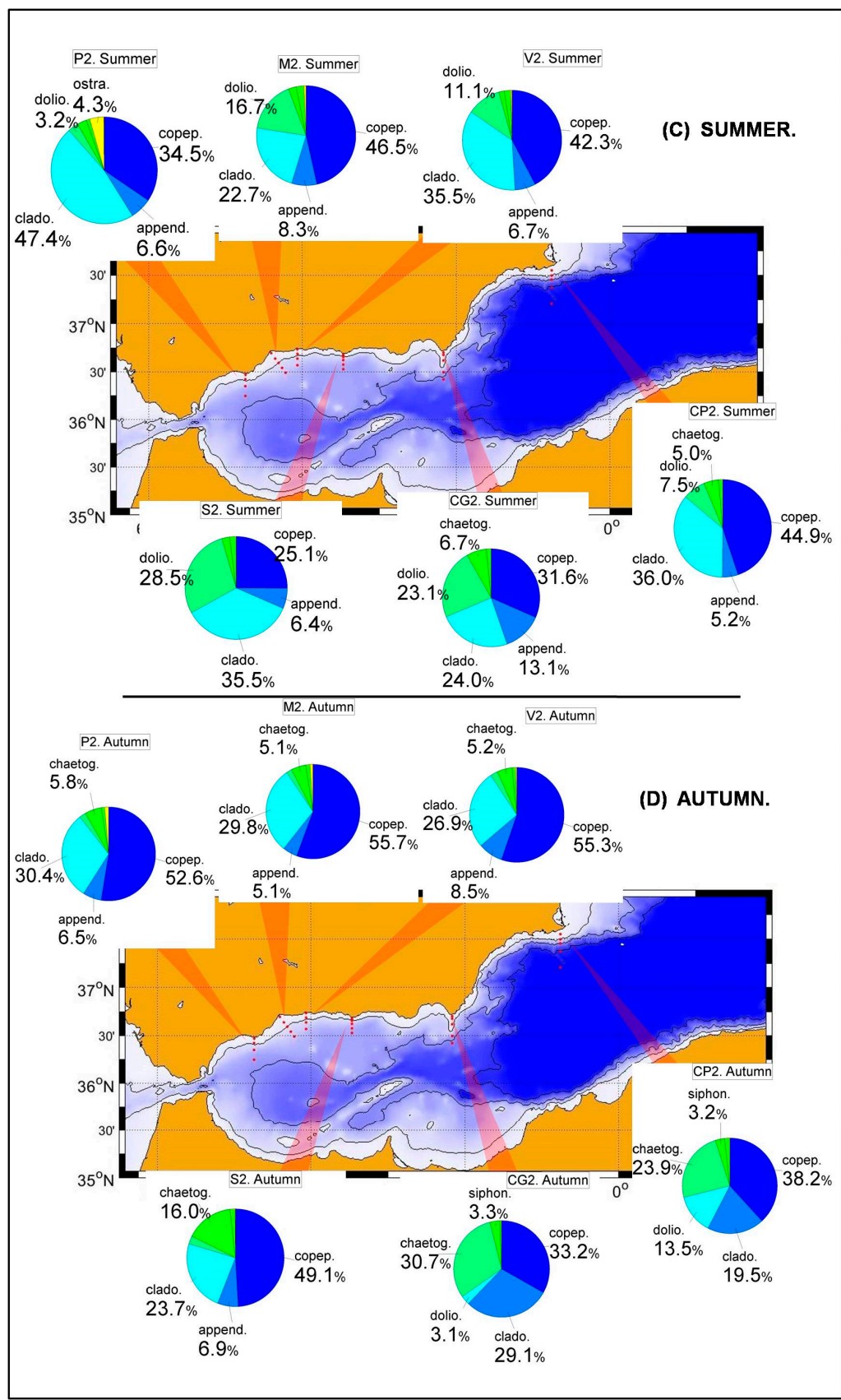

**Figure 10.** Relative abundances expressed as percentages for broad groups of the meso-zooplankton at the Alboran Sea and Cape Palos. (**A–D**) correspond to winter, spring, summer and autumn respectively.

Most of the stations to the north of Cape Palos do not allow us to present a complete seasonal cycle because of the numerous data gaps. Nevertheless, [43] presented the average abundances for the main meso-zooplanktonic groups from 1994 to 2003 at the same location occupied by the station B1, to the south of Mallorca Island. For comparison between the most productive waters in the Alboran Sea, and those more oligotrophic ones in the Balearic Sea [36], Figure 11 shows the seasonal cycle of the relative abundances for the main meso-zooplanktonic groups in B1 using results of Table 1 in [43]. Copepods were the most abundant group throughout the whole year with relative abundances higher than 50% for all the seasons. Appendicularians were the second most abundant group during winter, spring and autumn. In summer there was a considerable increase of the cladoceran relative abundance, as observed in the Alboran Sea, becoming the second most abundant group. Doliolids were present, although with low percentages throughout the year and chaetognaths appeared in summer and autumn.

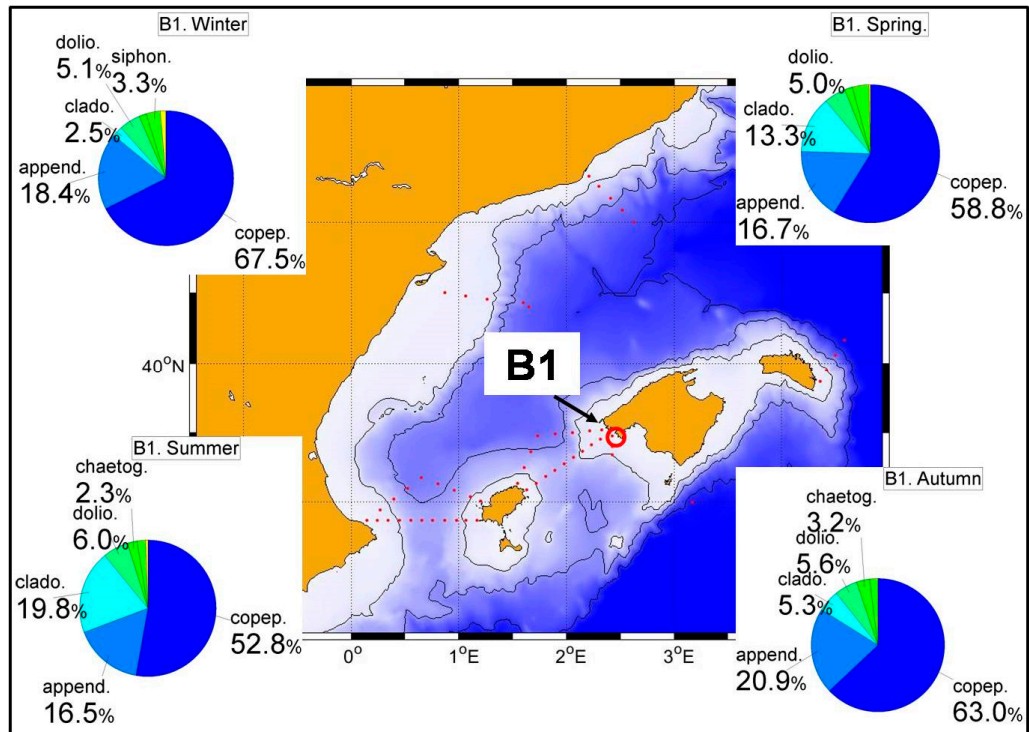

**Figure 11.** Relative abundances expressed as percentages for broad groups of the meso-zooplankton at the station B1 in Palma transect. This figure has been done using data in Fernandez de Puelles et al. [43].

Supplementary Materials Tables S9–S11 show the statistics for the main zooplanktonic group abundances in the Málaga region, while Table S12 presents the statistics for station B1 in the Mallorca transect using results in [43].

Figure 12 shows the mean average seasonal cycle for the meso-zooplanktonic biomass for the stations P2, M2 and V2 (left column) and the time evolution of the residuals (right column). Meso-zooplanktonic biomass in this area of the Alboran Sea oscillates between 10 and 30 mg/m$^3$. The maximum biomass seemed to occur in summer. Nevertheless, the four seasonal values were not statistically different at the 5% significance level (see confidence intervals for the mean values in Figure 12, left column). These time series were dominated by a period of high biomass between 1996 and 2003, mainly in P2 and V2 and less marked in M2. Very high values for some concrete years could reflect the existence of zooplanktonic blooms. Such values, although presented in the figures, were not considered for the calculation of the average seasonal cycles and trends. The linear trend for station P2 was not statistically significant, while those for M2 and V2 were significant with values of $-0.91 \pm 0.88$ mg m$^{-3}$/year and $-0.84 \pm 0.84$ mg m$^{-3}$/year respectively.

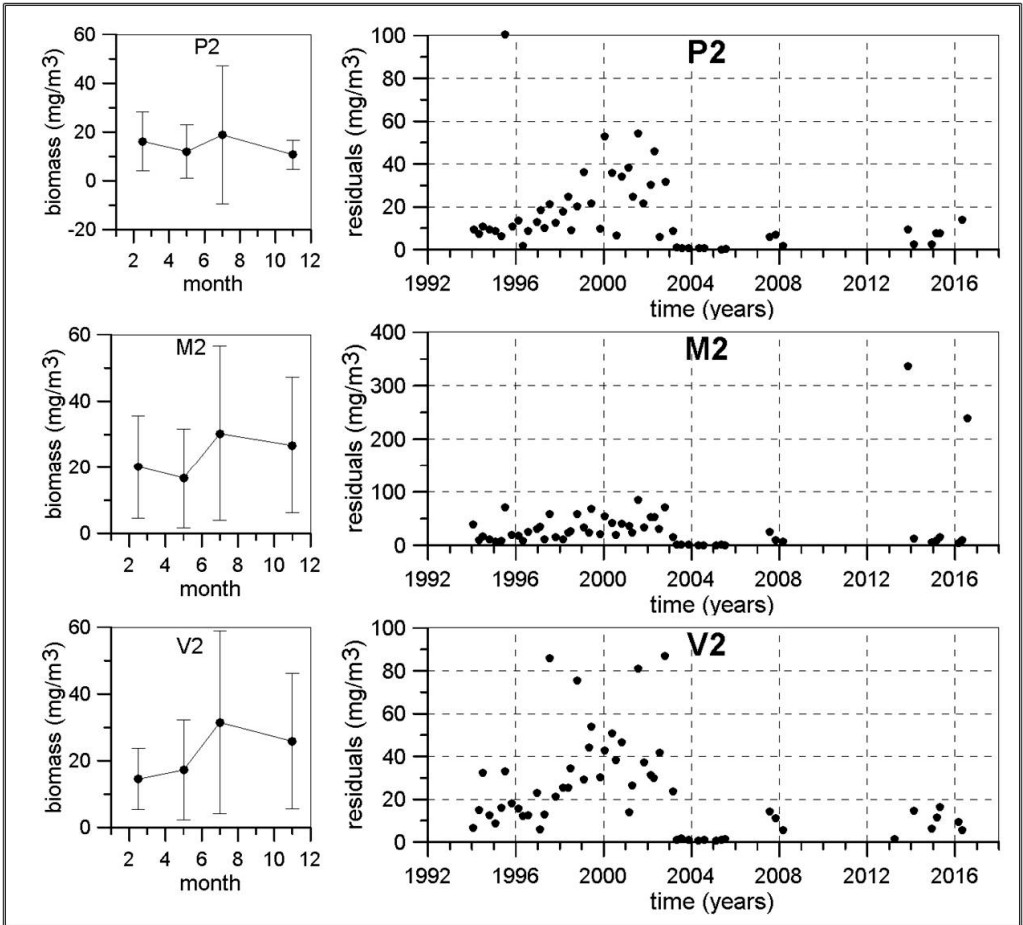

**Figure 12.** Left column shows the average seasonal cycle for meso-zooplanktonic abundances in mg/m$^3$ for the stations P2, M2 and V2 in the Alboran Sea. The right column shows the time series of residuals for the meso-zooplanktonic biomass at the stations P2, M2 and V2.

## 4. Discussion and Conclusions

According to the general picture usually accepted for the Mediterranean, it is an oligotrophic sea [44,45] where regenerated production prevails over the new one [24,46]. Intense cold and dry winds during winter produce the mixing of the upper water column supplying nutrients to the photic layer and producing a homogeneous vertical distribution of nutrients and phytoplankton abundance [47,48]. Our long-term seasonal patterns show that diatom abundances increase during winter or spring and this phytoplankton bloom is followed by a zooplankton bloom in spring-summer (highest zooplankton values in Figure 12 are observed in summer), which would agree with the increase of zooplanktonic abundances during summer previously reported [49]. In winter, maximum phytoplankton abundances can occur in the surface layer. As the spring advances, a seasonal thermo cline is developed, inhibiting the nutrient supply to the photic layer and nutrients are depleted by phytoplankton uptake [33] which in turn is grazed by zooplankton. During this stratified period, from mid-spring to late autumn, the phytoplankton is dominated by the picoplanktonic cells [50]. A deep chlorophyll maximum is developed and the depth of this maximum, which usually corresponds to the highest phytoplankton abundances, deepens from spring to autumn [36]. The zooplankton abundance also experiences a succession of groups. Copepods are the most abundant group in winter and spring, while cladocerans experience an important increase in summer becoming, in some cases, the most important group for the summer season.

Nevertheless, many works have shown that this general patternmay undergoimportant modifications as several mechanisms can fertilize the upper layer of the sea, altering the oligotrophy

of the Mediterranean waters and creating mesotrophic conditions [48,51]. Other works have shown that differentiated regions can be established in the Mediterranean Sea according to the concentrations of chlorophyll at the sea surface and along the water column,and the timing of the phytoplanktonic bloom [52,53]. Some of these differences have been evidenced within the RADMED area by means of the use of long time series of phyto and zooplankton data. One of the mechanisms that can inject nutrients to the photic layer, increasing the primary production and reducing the oligotrophy of the Mediterranean waters, is the existence of frontal systems and eddies [51,54,55]. The results of the present work show that diatoms are the most abundant group in the phytoplanktonic community in the westernmost stations of the Alboran Sea, not only in winter or spring, but throughout the whole year. From spring to autumn the maximum diatom abundances are observed at the sea surface or at the upper 20 m. It is also remarkable that summer abundances in the station P2 (Figure 2E) are higher than those corresponding to the winter or spring abundances in other regions of the RADMED area. As the highest abundances in the Western Alboran Sea occur in spring, this suggests that one of the causes for this phytoplankton bloom is the winter mixing, as in other areas of the Mediterranean Sea, and the spring prevalence of westerly winds which can induce wind-driven upwelling events in the northern coast of the Alboran Sea [56]. The persistence of high abundances of diatoms for most of the year suggests that other fertilizing mechanisms exist in this part of the Mediterranean. First, it has been hypothesized that the strong mixing in the Gibraltar Strait is a mechanism that acts throughout the year supplying nutrients to the upper layer of the sea [57,58]. These nutrient-rich waters would favor the phytoplankton growth at the northeastern side of the Strait and at the northwestern Alboran Sea. The high biomass produced by this mechanism would be advected by the Atlantic jet around the Western Alboran Gyre [59,60] affecting the transect P in the RADMED network. Besides this, two cyclonic circulation areas have frequently been described in the area between Gibraltar and Punta de Calaburras (to the west of transect P), and in front of Malaga Bay, affecting transects M and V, [54,61,62]. All these circulation traits do not seem to depend on the season and constitute permanent mechanisms for the fertilization of the photic layer, acting along the whole year. They would explain the high phytoplanktonic biomass in this sector of the Spanish Mediterranean with a weak seasonality if compared to other regions monitored in the RADMED project.

Concerning the nano and picoeukaryotes and the prokaryote pico-phytoplankton in the Alboran Sea, it is observed that *Synechococcus* abundances are maxima at the sea surface, decreasing with depth (Figure 2B,D,H). The only exception is the continental slope station P4 in summer, when the maximum abundances occur at 10 m depth. This behavior has frequently been observed, indicating the preference of this group for well illuminated waters and its adaptation to nutrient-poor waters [23,63,64]. By contrast, *Prochlorococcus* usually develop a deep maximum reflecting a better adaptation to low light intensities [63,64] permitting *Prochlorococcus* to take advantage, growing on deep nutriclines in stratified oligotrophic waters, not accessible for *Synechococcus*. Nevertheless, the observed *Prochlorococcus* maxima at 10 m depth in winter and spring and at 20 m in summer and autumn, suggest the existence of a shallow nutricline and a high nutrient availability in the uppermost layerof this part of the Alboran Sea, even during the stratified period.

Nutrient and chlorophyll distributions show the existence of a southwest-northeast gradient in the trophic conditions of the Spanish Mediterranean waters [36]. This is reflected in the phytoplanktonic distributions within the RADMED area. The prevalence of diatoms decreases eastwards within the Alboran Sea and to the north of Cape Gata, while the small flagellates are the most abundant group to the East of Cape Sacratiffor most of the year. Diatoms were the most abundant group in the Sacratif and Cape Gata transects only in winter, reflecting that the winter mixing is the main fertilizing mechanism in this area and the existence of a strong seasonality.

The aforementioned trophic gradient is accentuated to the north. The peninsular side of the Ibiza Channel and the transect to the south of Mallorca Island showed much lower micro-phytoplankton abundances (Figures 3 and 4A,C,E,G) than those observed in the Alboran Sea. The small flagellates group was the most abundant in both sites. In the station C20, within the Ibiza Channel, the diatoms

experienced an increase in spring when this group became the most abundant one only at the sea surface. *Synechococcus* showed maximum abundances at the sea surface or at the upper 25 m with the only exception of spring at stations C20 and C18 and summer at station C18 (Figure 3D,F), when the maxima were located at 50 m depth. *Prochlorococcus* showed deep maxima in all the cases, being these maxima at 25 m during winter, and deepening to 50 and even 75 m during spring, summer and autumn. Figures 3 and 4 show that nano and picoeukaryotes abundances ($\approx 10^3$ cel./mL) are an order of magnitude lower than those corresponding to the prokaryote group ($\approx 10^4$ cel./mL). This is another difference with the Alboran Sea where the picoeukaryotes have abundances comparable to those of the prokaryotes for some stations and seasons (Figure 2D,F,H).

In the northern sector of the RADMED area the conditions are similar to those observed in the Balearic Sea, although some differences can be outlined. Diatoms became the most abundant group in the micro-phytoplankton community in winter (Figure 5A), being the maximum values at 20 m depth. Although small flagellates were the main group in spring, high diatom abundances were observed at the sea surface (>80 cel./mL), and a deep maximum was developed between 50 and 75 m in summer (Figure 5B,C). If vertically integrated micro-phytoplankton abundances were considered, the dominance of diatoms was observed in the Barcelona transect during winter, and high diatom abundances were observed at Tarragona transect, also located in the Catalan Sea and to the south of the Barcelona transect. In all the other stations and seasons, the small flagellates were the most abundant group, with the only exception of autumn in station MH2 (northeast of Menorca Island). These patterns described above suggest that the main fertilizing mechanism easts of Cape of Gata is the winter mixing which is responsible for the increase of microplankton abundances. Nevertheless, there are also important differences within this area. It is well known that the Gulf of Lions and adjacent waters experience very strong vertical convection and a violent mixing of the water column in late winter [65]. These processes inject large amounts of nutrients to the photic layer enhancing primary production at late winter–early spring [47,48]. The intensity of winter winds and convection decreases southwards from the Gulf of Lions towards the Catalan Sea. This seems to be the cause for the higher winter micro-phytoplankton abundances at the Barcelona transect, where diatoms became the main group, and also of the importance of this group in Tarragona transect. As pointed out by [29] nanopankton seems to be less variable. Thus, the main differences between the different geographical areas are associated to the microplankton size fraction. Using satellite data, [52] classified the NW Mediterranean as a bloom site, that is, an area where an intense phytoplanktonic bloom occurs after severe winter mixing. The long-term time series of the RAMED project provides statistically robust data of the whole photic layer confirming that the Barcelona transect and, to a lesser extent the Tarragona one, belong to this category established by the cited authors.

The results from the statistical analysis presented in this work suggest that there are three main areas in the Spanish Mediterranean waters.First, the most productive waters located in the western part of the Alboran Sea where primary production and the presence of large cells such as diatoms are influenced by winter mixing, but also by other fertilizing processes linked to the dynamics of the Strait of Gibraltar [57,58] and the cyclonic circulation and frontal areas [32,33,66,67]. The consequence is a high phytoplanktonic biomass throughout the year. These high biomass values are also located at the very surface waters, indicating the nutrient supply to surface or sub-surface depths. Second, the poorest waters are those located to the south of the Balearic Islands and Cape Palos where small flagellates dominate during all the seasons. Third, the Catalan Sea, where the described oligotrophy could be partially "relaxed" by strong wind episodes, being winter mixing stronger than at the Balearic Sea. Figure S1 in Supplementary Materials offers a schematic description of these traits. In order to show more clearly the trophic gradient between the different RADMED regions, this figure is completed with nutrient and chlorophyll data from [36].

It is also important to mention that another factor able to fertilize the upper part of the water column within the Catalan-Balearic Sea is the frontal zone associated to the northern current flowing southward along the Catalan continental slope and then turning to the east at the northern continental

slope of the Balearic Islands (Balearic current, [68]). The only stations in the RADMED project that could be affected by this circulation feature are those at the transects BNA, T and MH. It is interesting to note that the station MH2 also showed a diatomdominance in summer (Figure 7C). Finally, dinoflagellates showed low abundances throughout the year with a summer increase in the stations of the Catalan-Balearic Sea in summer. This behavior had already been described and attributed to the ability of this group to adapt to low nutrient conditions because of the mixotrophy and mobility of numerous species within this group [69] allowing the migration between the lower limit of the photic layer and a deeper nutricline in summer. This summer increment in the relative importance of dinoflagellates is not observed in the Alboran Sea. We hypothesize that once again this is the result of the higher nutrient availability throughout the year where diatoms take advantage over dinoflagellates.

The length of some of the meso-zooplankton time series are greater than 20 years as this sampling was initiated in the early 1990s in the continental shelf of P, M and V transects. The other transects and the continental slope stations were included from 2007. Nevertheless, frequent problems in zooplankton sampling and the lack of personnel have not allowed us to construct long time series for meso-zooplankton abundances. Some of the few exceptions are the P2, M2 and V2 stations around Málaga Bay, where data from 1992 to 2000 from the previous monitoring program ECOMÁLAGA are available [70,71]. These data have been merged with abundances from years 2007, 2008 and winter 2009. The resulting statistics show that copepods are the main group in winter and spring in the whole Alboran Sea and Cape Palos. Notice that Sacratif, Cape Gataand Cape Palostransects were initiated in 2007 and because of the already mentioned problems, only one or two values are available for each season. In these cases the seasonal cycles obtained have been included for the completeness of the figures and they could be compared to other works dealing with just one single seasonal cycle, but they should be taken with caution within the frame of the present work.

Figures 10 and 11 show the dominance of copepods throughout most of the year with an important increase of cladocerans, and to a lesser extent, appendicularians in summer and autumn. Although the corresponding figures have not been shown and despite the low number of available samples, this behavior is common to all transects in the RADMED area. This succession of zooplanktonic groups coincides with previous works [21,22,55,71–73] and could be linked to the different feeding preferences. Copepods can select prey and tend to feed on large size particles from the micro-phytoplankton [74]. On the contrary, cladocerans are filters and mainly feed on pico and nanoplankton. Appendicularians would feed on the same size range than cladocerans [74,75]. Nevertheless, for the case of station P2 and other stations within the Alboran Sea, it cannot be established a clear decrement of large cells (diatoms) in summer, nor an increase of pico and nanoplankton that would justify the hypothesis mentioned above. In the case of the Balearic Sea, (Figures 3 and 4), and the Catalan Sea (Figure 5), the micro-phytoplankton abundance diminished during the summer season, supporting the meso-zooplankton groups succession associated to the change in prey availability.

Considering the total number of meso-zooplankton individuals and biomass (dry weight), there are also important differences from the south to the north. In the Alboran Sea the maximum values of total individuals are observed in summer with 1217 ind./m$^3$ at station P2 (Table S9) or 2172 ind./m$^3$ at the station V2 (Table S11). Biomass also reached a maximum in summer with 18.8, 30.2 and 31.5 mg/m$^3$ in P2, M2 and V2 respectively. By contrast, according to [43] (Table S12), the highest values of both abundances and biomass in station B1 were observed in spring with values of 1082 ind./m$^3$ and 6.4 mg/m$^3$. These figures simply reflect the lower productivity of the Balearic Sea which is transferred to the secondary producers.

Concerning the possible existence of long-term changes, the higher temperatures already observed in the Western Mediterranean [46] and in the RADMED area [15,36] could increase the thermal stratification of the water column decreasing the efficiency of winter mixing and the nutrient supply to the photic layer. This could lead to a negative trend in planktonic abundances and changes in the community structure with a higher dominance of small-sized cells which are expected to have

lower nutrient requirements [76–78]. The micro-phytoplankton and meso-zooplankton biomass time series in the Málaga Bay area (transects P, M and V) are the longest and more complete ones from the RADMED area. These time series extend over a 20 year-long period. If the strong inter-annual and decadal variability present in planktonic time series is considered, it should be concluded that 20 years is a short period of time, even more if the frequent data gaps are taken into account. For instance, [11] observed a negative trend for the chlorophyll-*a* concentrations at the station B at Villefranche from 1979 to 1998. This long-term change disappeared when the series was extended to 2005 [26]. All these reasons make us consider the present results with caution. Nevertheless, some negative trends have been estimated for the diatom abundances in the northwestern Alboran Sea and also for the meso-zooplanktonic biomass, which could be a signal of increasing stratification, less vertical nutrient supply, and decreasing micro-phytoplankton and zooplankton biomass. In summary, it can be concluded that the general oligotrophy of the Mediterranean waters is modulated in the Spanish Mediterranean by several mechanisms. In the Alboran Sea, the strong tidal mixing in the nearby Strait of Gibraltar and the existence of quasi-permanent frontal structures and cyclonic circulation cells are responsible for a high productivity which in turn produces the dominance of large phytoplanktonic cells (diatoms) throughout the whole year. Maximum values for such abundances are observed in the upper part of the water column indicating the availability of nutrients in the surface and sub-surface layers. The meso-zooplankton is dominated by copepods with an important increment of cladocerans and appendicularians in summer. The diatom abundances in this area are the highest of the whole RADMED area with values higher than 400 cel./mL. Zooplanktonic biomass also shows the highest values reaching more than 30 mg/m$^3$ in the summer season. The dominance of the diatom group disappears towards the eastern sector of the Alboran Sea, where small flagellates are the most abundant group. This trend continues and is accentuated to the north. The only exception is the winter season in the Barcelona and Tarragona transects which are affected by strong winter mixing. These processes would relax the oligotrophy of waters in the Catalan-Balearic Sea. Nevertheless, this effect decreases southward affecting in a much weaker way to the Ibiza Channel and Mallorca waters which seem to be the most oligotrophic ones in the RADMED area. The picoplanktonic fraction is the most abundant one in the phytoplanktonic community, with abundances of the order of 10$^4$ cel./mL, being followed by the nanoplanktonic fraction, which abundances are one order of magnitude lower (10$^3$ cel./mL). The general trend shows larger abundances for *Synechococcus* at the upper 20 m of the water column, while *Prochlorococcus* tend to develop a deep maximum from 25 m to 75 m. Nevertheless, this behavior shows frequent exceptions. Maximum values for *Synechococcus* abundances are at the sea surface and the deep *Prochlorococcus* maximum is at shallower waters in winter in most of the stations, when the nutrient supply to the photic layer is higher, and also in the Alboran Sea throughout the whole year. The zooplanktonic biomass in the Balearic Sea is lower than that observed in the Alboran Sea, reflecting the oligotrophy of these waters. Finally, the available time series do not allow us to obtain long-term trends for phytoplankton abundances and zooplanktonic biomass in most of the stations. Nevertheless, the best sampled area in the Málaga Bay, suggests a possible decrease of the large fraction of the phytoplankton (diatoms) and of the meso-zooplanktonic biomass. Such important questions should be followed in the future and evidence the importance of preserving monitoring programs and time series in the Mediterranean Sea and the world ocean.

**Supplementary Materials:** The following are available online at http://www.mdpi.com/2073-4441/11/3/534/s1, Figure S1: (A). Diagram showing the main traits of the RADMED area during the mixed layer period. Wind mixing enhances the nutrient supply into the photic layer. The legends inserted in the plot show for each area: the chlorophyll-*a* (green letters), nitrogen (nitrate plus nitrite, light brown letters) and phosphate (dark brown letters) concentrations integrated for the upper 100 m of the water column. Also in the legend are included the total number of micro_phytoplanktonic cells (diatoms plus dinoflagellates and small flagellates) and the most abundant micro-phytoplanktonic group. The four areas with differentiated characteristics are: Western Alboran Sea, Eastern Alboran Sea, Balearic Islands and Catalan Sea. These data show that the Western Alboran Sea is the most productive area, being the Balearic Islands the poorest one. Schematic profiles for the chlorophyll-*a* and the dissolved oxygen concentrations have been included showing that the highest values are observed in the upper part of the water column. Vertical blue dashed lines show the 4 and 5 mL/L dissolved oxygen concentrations. The

vertical green dashed line is the 1 mg/m$^3$ level and the brown vertical dashed line is the 1 μM concentration level. These values have been included to show more clearly the differences between the different areas and between the mixed layer period and the stratified period. All these data evidence the trophic gradient from southwest to northeast, with a slight increase of the productivity within the Catalan waters, if compared with the Balearic Islands. (B) Figure S1B. Diagram showing the main traits of the RADMED area during the stratified period. Concentrations of chlorophyll-*a* (green letters), nitrogen (nitrate plus nitrite, light brown letters) and phosphate (dark brown letters), integrated for the upper 100 m of the water column have been included. The total number of micro-phytoplanktonic cells (diatoms plus dinoflagellates and small flagellates) and the most abundant group are also included. These values are lower than during the mixed layer period. Nevertheless, this decrease is not so important in the western Alboran Sea as in the rest of the RADMED area. Mechanisms able to fertilize the photic layer throughout the whole year such as cyclonic circulation cells and fronts have been indicated. The position of the chlorophyll maximum in the Western Alboran Sea is in the upper 20 m of the water column and the dominant micro-phytoplanktonic group are the diatoms. On the contrary, a Deep Chlorophyll Maximum is developed to the east of the Alboran Sea and to the north of Cape Gata, being the small flagellates the most abundant group. The main zoopanktonic group is still the copepods, but there is an important increase of cladocerans and also appendicularians and doliolids. Table S1: Mean seasonal concentration (mean), standard deviation (σ), number de data used for calculations (n), minimum and maximum values reached in all the period (Min., Max.) for the main groups of microphytoplankton at station P2 at different depths. Table S2: Mean seasonal concentration (mean), standard deviation (σ), number de data used for calculations (n), minimum and maximum values reached in all the period (Min., Max.) for the main groups of microphytoplankton at station M2 at different depths. Table S3: Mean seasonal concentration (mean), standard deviation (σ), number de data used for calculations (n), minimum and maximum values reached in all the period (Min., Max.) for the main groups of microphytoplankton at station V2 at different depths. Table S4: Mean seasonal concentration (mean), standard deviation (σ), number de data used for calculations (n), minimum and maximum values reached in all the period (Min., Max.) for the main groups of microphytoplankton at station B1 at different depths. Table S5: Linear trends for the time series of residuals of diatom, dinoflagellate and small flagellate abundances at the stations P2, M2, V2 and B1. Confidence intervals at the 95 % confidence level have been included. Those trends which are statistically significant are in bold. Table S6: Mean seasonal concentration (mean), standard deviation (σ), number de data used for calculations (n), minimum and maximum values reached in all the period (Min., Max.) for the main groups of pico and nanoplankton at station P2 at different depths. Table S7: Mean seasonal concentration (mean), standard deviation (σ), number de data used for calculations (n), minimum and maximum values reached in all the period (Min., Max.) for the main groups of pico and nanoplankton at station B2 at different depths. Table S8: Mean seasonal concentration (mean), standard deviation (σ), number de data used for calculations (n), minimum and maximum values reached in all the period (Min., Max.) for the main groups of pico and nanoplankton at station BNA2 at different depths. Table S9: Mean seasonal zooplankton abundance (mean), standard deviation (σ), number de data used for calculations (n), minimum and maximum values reached in all the period (Min., Max.) for the main groups of zooplankton at station P2. Table S10: Mean seasonal zooplankton abundance (mean), standard deviation (σ), number de data used for calculations (n), minimum and maximum values reached in all the period (Min., Max.) for the main groups of zooplankton at station M2. Table S11: Mean seasonal zooplankton abundance (mean), standard deviation (σ), number de data used for calculations (n), minimum and maximum values reached in all the period (Min., Max.) for the main groups of zooplankton at station V2. Table S12: Mean seasonal zooplankton abundance (mean), standard deviation (σ), number de data used for calculations (n), minimum and maximum values reached in all the period (Min., Max.) for the main groups of zooplankton at station B1. From Fernández de Puelles et al. (2007), [43].

**Author Contributions:** Conceptualization, M.d.C.G.-M. and M.V.-Y.; methodology, M.d.C.G.-M., R.S., and F.M.; formal analysis, M.V. and M.d.C.G.-M.; investigation, M.d.C.G.-M., M.V.-Y. and A.R.; data curation, F.M. and M.M.; writing—original draft preparation M.V.-Y, M.d.C.G.-M.; writing—review and editing, M.V.-Y., A.R., M.d.C.G.-M. and R.B.; project administration, R.B.; funding acquisition, J.L.L-J.

**Funding:** The RADMED monitoring program is funded by the Instituto Español de Oceanografía, and has been partially funded by the DESMMON project (PN I+D+I CTM2008-05695-C02-01), the PERSEUS project (FP7-287600), the IRIS-SES project (DG ENV GA-07.0335/2013/659540/SUB/C2.), the ActionMed project (DG-ENVGA-11.0661/2015/12631/SUB/ENVC.2) ATHAPOC project (PN I+D+I CTM2014-54374-R).

**Conflicts of Interest:** The authors declare no conflict of interest.

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
