# Peer review of "Spatial and Temporal Long-Term Patterns of Phyto and Zooplankton in the W-Mediterranean: RADMED Project"

_water, doi:10.3390/w11030534_

Round 1

Reviewer 1 Report

Good work here. I find the significance of your work to be high and well worth revising your article for final publication. Please understand that I appreciated the structure and flow of your manuscript, the core ideas, the effort to evaluate the best available data time series on multiple water quality parameters, and the effort to tie together the story behind the results. 

There is a lot of good work presented here. As I worked my way through the review I made fairly substantive notes to provide you with recommendations on improving the text and presentation. For this reason I suggest the manuscript be accepted, and the reference to major revisions here reflects a few places where 1) the text deserves clarification at a paragraph scale, 2) a suggestion that you could consider strengthening the presentation by including a conceptual diagram of the forcing functions on your plankton communities because so much of the paper talks presents pieces of the cycle and one diagram could be a strong single reference for each seasonal argument making linkages between plankton and nutrient dynamics (i.e., showing the seasonal cycle of what influences nutrient availability, what are the sources of nutrients and where are they derived, and in what seasons you have water column structure influencing access to the nutrients including what climate drivers influence these conditions in each season, e.g. more wind, more mixing, low anthropogenic inputs) and 3) the graphics are very nicely presented, however, you can improve the presentation b, for example, condensing individual seasonal pie chart information into a stacked bar format and produce 1 instead of 4 figures (e.g. Figs 8,9,10,11) and residual trend graphics that can show data and trend line (instead of the disconnected lines in the graphics now), reducing them in size to fit on one page, and adding a table of regression results.

And please note, all figures and tables need stand-alone information about what is contained in them so that they may be understood independently by the reader. A figure or table title that states "Same as the previous one but for station X" tells me nothing about the content of that figure or table presentation. Take time to give the reader the understanding they need with each figure or table presented in the article.  

Continue with your good work. Good luck! 

Best Regards, 
Your Reviewer

Water – Review. Sharing my notes. 

Line 101: “The goal of the present work is to complete the previous ones…”

I don’t understand what it means to have a goal to “complete the previous ones…”.  What I see is that you have a data time series. You are assembling the data. You are characterizing the data. You definitely have a goal of analyzing time series for spatial and temporal patterns in phytoplankton and zooplankton abundance and distributions from the RAD*** program database.

Line 102: “…and provide baseline data that could be used for regionalization of the pelagic ecosystem, initialization or validation of ecological models or simply as a reference for future works.”

I find it confusing to call the work “regionalization of the pelagic ecosystem”. I don’t understand what this phrase means. It seems that what you want to do with your work and presentation here is to consolidate, verify, and make available baseline data to support water quality and aquatic living resource characterization of the western Mediterranean Sea ecosystem. The datasets can further support development and validation of ecological and management-decision models of the ecosystem.

Line 108: (ii) The average cycles were estimated for the abundances of the nano and pico‐eukaryote…

I don’t know what an “average cycle” is in the way the words are used here. Is this the average time between peak abundances in a day, a week, a month, a year, over multiple years? What pattern in their abundances defines “a cycle”? What are the criteria for the definition of a cycle? What is being averaged?

Line 110: “Decadal changes were studied at the stations where the time series extension made it  possible.”

Please include reference to decadal changes of what it is being evaluated. For example, you might mean that “Decadal changes in phytoplankton and zooplankton density, biomass, taxonomic composition and spatial distributions were evaluated at a subset of monitoring stations where multi-decadal time series data were available.”

Line 112: “The seasonal cycle of zooplanktonic biomass and the abundances of broad zooplanktonic groups were obtained.

Do you mean “the seasonal cycle­s of zooplankton biomass and the abundances of broad zooplankton taxonomic groups were obtained. (?)

OR

Do you mean “the seasonal cycle of mesozooplankton biomass were characterized across the Mediterranean Sea study region. Further, the variability in zooplankton density within and between years for dominant groups and species were summarized.” (?)

Be specific. My use of meso here was being specific to larger sizes per the net size used in the sampling program. However, this text about seasonal cycle assessment in reference to all zooplankton in the study, or just micro, or just meso, or micro, meso and macrozooplankton? Please guide the reader and researcher that will be reading your article a bit more here with details to provide a clearer, more specific picture of what they are about to see in the results. 

Lines 115-121: Recommending rephrasing this paragraph.

 obtained can be considered as based on robust statistics. In such cases, decadal changes can also be  estimated. At other stations, only a complete seasonal cycle (one data per season) is available, and results should be considered with caution. In these latter cases the results are presented for the completeness of the work and are discussed to the light of previous works also dealing with four cruises distributed along one single year. Section 2 is a detailed description of the sampling and analysis strategy within the RADMED project. Main results are in section 3 and a discussion and conclusions are presented in section 4.

Line 142. CTD – define this abbreviation the first time you use it.

Line 143.  SBE – define this abbreviation the first time you use it.

Line 149. What do you mean by this “For the triangles around the Balearic Channels…”. Triangles? What triangles? I am sorry, but I don’t understand this sentence as it is presented.

Line 150. Is there any analysis behind choosing the two stations as representative of the associated habitat conditions? Is there some evidence that water quality test results tend to be average for the region?

Line 155. Sampling at “stations 2…”. Do you mean sampling at a station that is identified as #2, or do you mean “2 stations” as in 2 different locations? Based on line 157 I think you mean “station 2 (continental shelf)” here. 

·         Note: Something helpful for the reader besides the figure could be a supplemental file that identifies all the stations codes you are using and their associated, spelled out definitions. This is an appropriate reference example because “station 2” means nothing to me. “Station 2 (continental shelf)” still means nothing to me. But if I could refer to the figure and a supplemental table that has a simple definition of the location (e.g., Station 2 – continental shelf – Malaga transect (or whatever the appropriate geographic identifier is), it would allow me to understand the context of your reference to this site.  I believe it would be effective to have this available for all readers of your article.

Line 160. Samples are stored in darkness at ambient temperatures? (By comparison, I see you reference the storage temperature for pico and nano plankton samples in line 166-167. Therefore, a reference to temperature associated with storage conditions here is complementary to explaining the treatment of the samples.)

Line 194. weighted? I believe you mean “mass” of the filter was recorded.

 Line 196-197. “weight” should be “mass” here.

Line 199-202.  I think what you mean here is that “the purpose of the project is to define baseline environmental conditions to serve as a reference to understanding change over time. Baseline metrics of the present ecosystem status include annual averages, seasonal averages, annual variability, seasonal variability” (and maybe more, but I think you have two sentences here out of the one.)

Line 202-205. Rephrase please.

The first one is that they should provide some insight about the environmental state of the Spanish Mediterranean waters, and second, that they should be easily monitored with a long term perspective consuming the less possible personnel and vessel time.

The highlighted text might be summarized by saying something like your monitoring program is aiming for maximizing data acquisition while optimizing the economic efficiency of the monitoring program.

Line 205. End of sentence:  analyzes, (not “analyses”).

Line 207-209: Instead of your language in this paragraph that begins with, “For each depth, mean values are estimated for each season. Standard deviations and the minimum and maximum values recorded are also obtained for characterizing the time variability associated to each variable and season”, what you describe are elements of a univariate statistical summary of the data. A common way of saying this more concisely is  “Univariate statistics were computed for bacteria and mesozooplankton abundances by depth within seasons for each year.”  (or if you combined years at each depth you would not say ‘for each year’).

Line 220. “personal” should be “personnel”.

Line 235. Naoplankton should be nanoplankton.  

Table 1. Check that information in certain columns is cutoff/incomplete visually within the information cell in the table. This was especially noticeable for information in the first column.

Line 254. Donoflagellates should be dinoflagellates

Confusing mixing and matching of references to the same transects: Capre Gata vs. Cape Gata vs. CG.

Figure 8, 9, 10, 11 –

First - why do the lines connect some data no matter how much time passed between them while not connecting other segments of the graphs. A more effective presentation would be to remove the lines connecting the data, then add in the trend line with confidence intervals.

Second - Give complete figure descriptions to each figure. Imagine a reader might extract a single graphic to reference elsewhere. A figure description that says, for example in Fig 9 “Same as Fig 8 but for M2” says nothing about what information is actually being portrayed about plankton community group time series for Alboran Sea station (M2) and the trends on their residuals from a linear trend analysis, trend significance assessed at P<0.05. (Or, similar language. This is a just suggested form of text that should be associated with each figure.). Pay attention to this for all your graphics in the document. There is another solution to help this issue along per my third suggestion.

Third – If you advance to graphics with data and trend lines, you should be able to shrink the size of each graph and potentially get all of them on 1 page. Alternatively, you could produce a single graph example in the main body of the text here and then provide a supplemental file of the remaining graphics.

Finally – if you have regression analyses and significance tests for all these data, a common summary table of the regression equation and significance test results (p-values) allows a more concise summary to the reader than noting slopes for each individual assessment in the text. Your text then can focus up one level from individual slopes to describe the range of slopes of change in abundance residuals for example and the average of the slopes, etc.

Throughout the document, I would suggest spelling out “small flagellates” rather than some places using the text small flagellates and other places using “small flagel” as some mysterious abbreviation of the full title of the group.

Line 393 18.3 not 18,3

Line 394 Update punctuation in the sentence a little here. One approach is to use brackets, such as “The remaining groups (doliolids,…,sychophysans) were less abundant…”.  Alternatively, you could get rid of the use of brackets altogether by eliminating the phrase “The remaining groups” and just state “The dolioids,…, and sychophysans were less abundant…”

Figure 12. I appreciate the single season focus of the graphics. The presentation is nice. However, I suggest the information of four seasonal graphics should be condensed into one here. I recommend using the same base picture of the sampling region, but instead of 4 seasonal pie charts on 4 separate figures, produce a stacked bar chart for each location with 4 bars representing each of the 4 seasons for a location, and the segments of each stack represent the % of the taxonomic groups counted within each season. Such a presentation should work well for you and obviously save you space.

I am unfamiliar with the abbreviation “cel./ml”.  I have always seen this spelled out as cells/ml (as you did in line 630) or cells*mil-1. I would say the same for your use of ind./ml. I defer to the editor for a final decision about which presentation of these units is most appropriate for this journal.

Line 448. “Stormy winter activity…” – I don’t know what this means. Do you mean wind storms, sand storms, rain storms, snow storms? Or do you mean something like “Winter weather, with its associated long durations of high wind speeds and increased precipitation to the region compared to average spring, summer and autumn weather conditions,…”, or do you mean something else? Please be more descriptive and specific here. I think your text around line 540 gets at the type of translation for “stormy conditions” that would be helpful here. Thank you.

Line 454-456. This information would be more valuable to the article if you lead into it with a couple of sentences that define the conceptual model for nutrient loading and availability in the water column for the Mediterranean Sea. 

You state:

“In winter, maximum phytoplankton abundances can occur in the surface layer. As the spring advances, a seasonal thermocline is developed, inhibiting the nutrient supply to the photic layer and nutrients are depleted by phytoplankton uptake [30] which in turn is grazed by zooplankton.”

In other temperate ecosystems, spring is a time of increased rainfall. Elevated rainfall increases the nutrient delivery to the main body of a sea or estuary. Fresher water moves out over the saltier water feeding the phytoplankton in the photic zone. Your statements here suggest the Mediterranean Sea productivity is dominated by the deep water delivery of nutrients to the photic zone when there are well mixed conditions throughout the water column, and that nutrient loading from the land and air is insufficient to significantly influence plankton production during portions of the year where the water column has structure (i.e., thermocline at minimum).  I believe that some brief text then that gives the reader this sort of overview of the annual water quality dynamics over an annual cycle relating seasonal nutrient loading sources with nutrient availability, contrasting a minimum impact of land-and-air based nutrient sources and the dominance of ocean-based nutrient delivery, feeding plankton production of the sea as understood in this point in time will bridge the excellent ideas you are sharing with an understanding about how your ecosystem behaves as important context to support your argument. Just a little up front presentation of the system appears to me as the link many of your discussion elements can tie back to throughout your article. You may even consider providing such a conceptual diagram of the seasonal nutrient availability and loading sources in your introductory material as a firm basis for how you link your findings back to the understanding about the best understanding for how the ecosystem functions at this time.

Line 460-463. Is the seasonal succession of zooplankton species typical or unusual for inland seas? Can you please provide supporting references that give context to this statement as to whether this is a normal or abnormal successional progression with respect to zooplankton ecology? Otherwise, the statement is a result that belongs in the results section and not much of a discussion issue here.

Line 471. What does it mean to “relax oligotrophy”? Do you mean there is ” a hydrodynamic process that can enhance nutrient availability in the photic that results in elevated plankton production”?  

Line 490. Given the impacts of climate change on slowing certain ocean circulation patterns like the Gulf Stream in the Atlantic Ocean for example, I would caution you to describe any circulation feature as “permanent”. There may be limited evidence of change in these circulation features for the Mediterranean Sea at this time, however, climate change effects may impact the functioning of such processes in the future. (I see you actually reference climate change effects on water column behavior in line 612.) Perhaps you have worked with others that have produced model scenarios of increasing temperatures, the impact of more freshwater in the Atlantic from ice melt, projected changes in precipitation patterns to your region, etc. that might suggest there is no evidence circulation features will be impacted significantly by climate change in the near future (e.g., 50-100 year projections). That would be helpful to reference. In the meantime, consider using a different term than “permanent”.

Line 551. A single sentence as a paragraph is rather unusual for an article. You have several in succession here. I believe what you want here is something restructured into a single paragraph, more like this:

The results from the statistical analysis presented in this work suggest that there are three main areas in the Spanish Mediterranean waters.  First, the most productive waters located in the western part of the Alboran Sea where primary production and the presence of large cells such as diatoms are influenced by winter mixing, but also by other fertilizing processes linked to the dynamics of the Strait of Gibraltar [52,61] and the cyclonic circulation areas [62,30]. The consequence is a high phytoplanktonic biomass throughout the year. These high biomass values are also located at the very surface waters, indicating the nutrient supply to surface or sub‐surface depths. Second, the poorest waters are those located to the south of the Balearic Islands and Cape Palos where small flagellates dominate during all the seasons. And third, the Catalan Sea, where the described oligotrophy could be partially “relaxed” by strong wind episodes, being winter mixing stronger than at the Balearic Sea.

Line 577. For this paragraph, I would recommend starting the paragraph emphasizing the complete long time series you were able to work with first, then refer to the abundance of limited data sets, and end the paragraph similar to what you have done with suggestions for better continuity throughout the region to enhance understanding of zooplankton dynamics.

Line 584, 585 – no need for identifying the abbreviations again at this point. That should be accomplished early in the article.

Line 630. “Cuasi-“ should be “quasi-“

Final sentence. Please rephrase this sentence. It is choppy in its presentation.

Below – this document should be a useful reference to include in your article.

Nutrient Cycling in the Mediterranean Sea: The Key to Understanding How the Unique Marine Ecosystem Functions and Responds to Anthropogenic Pressures.

By Helen R. Powley, Philippe Van Cappellen and Michael D. Krom

Submitted: December 2nd 2016 Reviewed: September 7th 2017 Published: November 8th 2017. DOI: 10.5772/intechopen.70878

Author Response

Good work here. I find the significance of your work to be high and well worth revising your article for final publication. Please understand that I appreciated the structure and flow of your manuscript, the core ideas, the effort to evaluate the best available data time series on multiple water quality parameters, and the effort to tie together the story behind the results. 

There is a lot of good work presented here. As I worked my way through the review I made fairly substantive notes to provide you with recommendations on improving the text and presentation. For this reason I suggest the manuscript be accepted, and the reference to major revisions here reflects a few places where 1) the text deserves clarification at a paragraph scale, 2) a suggestion that you could consider strengthening the presentation by including a conceptual diagram of the forcing functions on your plankton communities because so much of the paper talks presents pieces of the cycle and one diagram could be a strong single reference for each seasonal argument making linkages between plankton and nutrient dynamics (i.e., showing the seasonal cycle of what influences nutrient availability, what are the sources of nutrients and where are they derived, and in what seasons you have water column structure influencing access to the nutrients including what climate drivers influence these conditions in each season, e.g. more wind, more mixing, low anthropogenic inputs) and 3) the graphics are very nicely presented, however, you can improve the presentation b, for example, condensing individual seasonal pie chart information into a stacked bar format and produce 1 instead of 4 figures (e.g. Figs 8,9,10,11) and residual trend graphics that can show data and trend line (instead of the disconnected lines in the graphics now), reducing them in size to fit on one page, and adding a table of regression results.

And please note, all figures and tables need stand-alone information about what is contained in them so that they may be understood independently by the reader. A figure or table title that states "Same as the previous one but for station X" tells me nothing about the content of that figure or table presentation. Take time to give the reader the understanding they need with each figure or table presented in the article.  

Continue with your good work. Good luck! 

Best Regards, 
Your Reviewer

First of all we would like very sincerely to thank the reviewer for his/her very exhaustive and constructive review.

1) The reviewer has provided some general comments above this lines. We have followed most of these comments and suggestions. The answers to this general points are the first ones that we present in this document.

2) The reviewer has provided many interesting detailed comments (in bold), later in this document. We have followed almost all of them and provided answers to all the comments. Such answers are inserted after each comment. We sincerely think that the new version is considerably improved thanks to the reviewer's comments and suggestions.

1) General comments.

First of all the reviewer suggests that the manuscript needs clarification at a paragraph scale. Certainly we agree with the reviewer. In fact, he/she has provided many suggestions. We have followed most of them. We submit a new version with the changes highlighted. Some of these corrections are explained in the following answers. We hope that now the manuscript clearer.

Second, the reviewer suggests the inclusion of a conceptual diagram in the supplementary material explaining the main forcing mechanisms acting on the planktonic community. We have followed this suggestion and included Fig.S1. This figure has been completed with data obtained from [34]. Please notice that the second reviewer had asked to include some references that supported the trophic gradient of the RADMED area with nutrient and chlorophyll data. This diagram and the use of the information from [34] give answer to both suggestions.

Third, the reviewer suggests the possibility of reducing the number of figures integrating different figures corresponding to different seasons into one single figure. The reviewer also suggest to eliminate the lines connecting dots in figures 8, 9, 10 and 11 and simply inserting the trend lines. Somewhere along this document the reviewer also suggests to include a table with the linear trends. He/she also suggests that all the figures should have their own legend.

First we have included a legend for each figure instead of using "the same as..." that was used in the previous manuscript. We have also eliminated lines connecting dots in figures 8 to 11 and merged these four figures into two new figures: 8 and 9 in the new version. Lines representing the linear trends have been included. We have also provided a table in supplementary material (because of the length of the table) with all the trends estimated for stations P2, M2, V2 and B1 and for all the depths analyzed. This new table is table S5.

Water – Review. Sharing my notes. 

 Line 101: “The goal of the present work is to complete the previous ones…”

I don’t understand what it means to have a goal to “complete the previous ones…”.  What I see is that you have a data time series. You are assembling the data. You are characterizing the data. You definitely have a goal of analyzing time series for spatial and temporal patterns in phytoplankton and zooplankton abundance and distributions from the RAD*** program database.

Ok, the phrase " to complete previous ones" has been suppressed.

Line 102: “…and provide baseline data that could be used for regionalization of the pelagic ecosystem, initialization or validation of ecological models or simply as a reference for future works.”

I find it confusing to call the work “regionalization of the pelagic ecosystem”. I don’t understand what this phrase means. It seems that what you want to do with your work and presentation here is to consolidate, verify, and make available baseline data to support water quality and aquatic living resource characterization of the western Mediterranean Sea ecosystem. The datasets can further support development and validation of ecological and management-decision models of the ecosystem.

Ok! We accept that the redaction proposed by the reviewer is clearer than ours. We have changed this paragraph following his/her suggestion. The only objection is that we have changed "Mediterranean ecosystem" by Spanish Mediterranean ecosystem, as our monitoring area is reduced to the Spanish waters.

Line 108: (ii) The average cycles were estimated for the abundances of the nano and picoeukaryote…

I don’t know what an “average cycle” is in the way the words are used here. Is this the average time between peak abundances in a day, a week, a month, a year, over multiple years? What pattern in their abundances defines “a cycle”? What are the criteria for the definition of a cycle? What is being averaged?

 The reviewer is right. Maybe this needs a more clear redaction. The average cycle for any property is obtained taking all the measurements or data corresponding to each season and then estimating the mean value and the standard deviation for that particular season. Repeating this procedure for the four seasons we obtained what could be considered as an average seasonal cycle. Obviously each year will be different and can depart from this average cycle. In the case of some properties such as temperature and salinity or nutrients, the usual term is a "climatological seasonal cycle". nevertheless, the term climatology implies that time series are at least 30 years long. For this reason we find more appropriate the term average seasonal cycle. The four mean values for each season could show the differences between seasons that are more frequently observed. It is also true that estimating monthly average values would define more accurately the seasonal cycle, but in this case the sampling is seasonal.

We have changed the redaction of this paragraph to make it more clear and to explain to the reader what the average seasonal cycles are considered in the present work.

Line 110: “Decadal changes were studied at the stations where the time series extension made it  possible.”

Please include reference to decadal changes of what it is being evaluated. For example, you might mean that “Decadal changes in phytoplankton and zooplankton density, biomass, taxonomic composition and spatial distributions were evaluated at a subset of monitoring stations where multi-decadal time series data were available.”

 Ok! the reviewer is right. The existence of decadal changes were analyzed for the time series of micro-phytoplanktonic abundances and for the meso-zooplanktionic biomass in some stations. This paragraph has been re-written to make it clearer.

Line 112: “The seasonal cycle of zooplanktonic biomass and the abundances of broad zooplanktonic groups were obtained.

Do you mean “the seasonal cycle­s ofzooplankton biomass and the abundances of broad zooplankton taxonomic groups were obtained. (?)

OR

Do you mean “the seasonal cycle of mesozooplankton biomass were characterized across the Mediterranean Sea study region. Further, the variability in zooplankton density within and between years for dominant groups and species were summarized.” (?)

The first option is the right one. We have corrected it in the new version.

Be specific. My use of meso here was being specific to larger sizes per the net size used in the sampling program. However, this text about seasonal cycle assessment in reference to all zooplankton in the study, or just micro, or just meso, or micro, meso and macrozooplankton? Please guide the reader and researcher that will be reading your article a bit more here with details to provide a clearer, more specific picture of what they are about to see in the results. 

 The reviewer is right. In the case of phytoplankton, the three usual size ranges that are sampled are: micro, nano and pico plankton. Their size ranges have been specified in this paragraph. As the reviewer suggests, this offers the readers a clearer picture of what they are going to find.

Lines 115-121: Recommending rephrasing this paragraph.

 obtained can be considered as based on robust statistics. In such cases, decadal changes can also be  estimated. At other stations, only a complete seasonal cycle (one data per season) is available, and results should be considered with caution. In these latter cases the results are presented for the completeness of the work and are discussed to the light of previous works also dealing with four cruises distributed along one single year. Section 2 is a detailed description of the sampling and analysis strategy within the RADMED project. Main results are in section 3 and a discussion and conclusions are presented in section 4.

Ok! We have rephrased this paragraph, trying to explain which kind of statistics have been estimated as also suggested by the reviewer in other comment. 

Line 142. CTD – define this abbreviation the first time you use it.

Ok! The definition has been included.

Line 143.  SBE – define this abbreviation the first time you use it.

SBE is the Sea-Bird Scientific company. This has been included in the new version.

Line 149. What do you mean by this “For the triangles around the Balearic Channels…”. Triangles? What triangles? I am sorry, but I don’t understand this sentence as it is presented.

We have modified the redaction to make it clear that the oceanographic stations in the Ibiza and Mallorca Channels form two triangles. We have also include a reference to figure 1 where these triangles can be seen clearly and where the positions of stations C20 and C18 are included.

Line 150. Is there any analysis behind choosing the two stations as representative of the associated habitat conditions? Is there some evidence that water quality test results tend to be average for the region?

The reason to choose stations C20 and C18 is the following one. If only the peninsular coast is considered, the RADMED program monitors the continental shelf and the continental slope from Malaga, in the southwestern part of the coast, to Barcelona, at the northeastern sector. In the Ibiza Channel we simply try to complete the monitoring of the peninsular coast. Therefore we choose C20 because it is on the shelf (depth lower than 200 m), and C18 because it is on the continental slope. In this way the shelf/slope monitoring has a continuity along the coast. The islands have a different treatment for logistic reasons. The coast is much more complex. For this reason we include the B transect for historical reasons, as it was initiated under the umbrella of a local project in 1994. As this transect is to the south of the islands, we also included the transect MH to the north of Menorca. This latter transect is affected by the Balearic Current.

Line 155. Sampling at “stations 2…”. Do you mean sampling at a station that is identified as #2, or do you mean “2 stations” as in 2 different locations? Based on line 157 I think you mean “station 2 (continental shelf)” here. 

We accept that this could be confusing. We have changed the redaction using the second and forth station of each transect as suggested by the reviewer.

·         Note: Something helpful for the reader besides the figure could be a supplemental file that identifies all the stations codes you are using and their associated, spelled out definitions. This is an appropriate reference example because “station 2” means nothing to me. “Station 2 (continental shelf)” still means nothing to me. But if I could refer to the figure and a supplemental table that has a simple definition of the location (e.g., Station 2 – continental shelf – Malaga transect (or whatever the appropriate geographic identifier is), it would allow me to understand the context of your reference to this site.  I believe it would be effective to have this available for all readers of your article.

We think that this is clear with the map in figure 1 and the changes made in the text, substituting the stations 2 and 4 by the second and the fourth for each transect. We think that now the reader can have a clear idea about the station considered for each case.

Line 160. Samples are stored in darkness at ambient temperatures? (By comparison, I see you reference the storage temperature for pico and nano plankton samples in line 166-167. Therefore, a reference to temperature associated with storage conditions here is complementary to explaining the treatment of the samples.)

This samples are stored at ambient temperature. As suggested we have included this information at line 191.

Line 194. weighted? I believe you mean “mass” of the filter was recorded.

Ok! We have corrected this.

 Line 196-197. “weight” should be “mass” here.

The reviewer is right. Strictly speaking it is a mass, not a weight, which is a force and should be expressed in Newton. We have corrected this and changed weight by mass as suggested.

Line 199-202.  I think what you mean here is that “the purpose of the project is to define baseline environmental conditions to serve as a reference to understanding change over time. Baseline metrics of the present ecosystem status include annual averages, seasonal averages, annual variability, seasonal variability” (and maybe more, but I think you have two sentences here out of the one.)

Line 202-205. Rephrase please.

The first one is that they should provide some insight about the environmental state of the Spanish Mediterranean waters, and second, that they should be easily monitored with a long term perspective consuming the less possible personnel and vessel time.

The highlighted text might be summarized by saying something like your monitoring program is aiming for maximizing data acquisition while optimizing the economic efficiency of the monitoring program.

Ok! We have included the sentence proposed by the reviewer: " is aiming for maximizing data acquisition while optimizing the economic efficiency of the monitoring program" as we accept that it better summarizes the meaning of this paragraph. We have changed other sentences in the previous lines to make clearer the meaning of this paragraph.

Line 205. End of sentence:  analyzes, (not “analyses”).

 Ok!, corrected when it is a verb.

Line 207-209: Instead of your language in this paragraph that begins with, “For each depth, mean values are estimated for each season. Standard deviations and the minimum and maximum values recorded are also obtained for characterizing the time variability associated to each variable and season”, what you describe are elements of a univariate statistical summary of the data. A common way of saying this more concisely is  “Univariate statistics were computed for bacteria and mesozooplankton abundances by depth within seasons for each year.”  (or if you combined years at each depth you would not say ‘for each year’).

 In this case we prefer to keep the present redaction because univariate statistics is not very specific. Different kinds of analyses can be carried out and we prefer to specify which ones are presented in this work.

Line 220. “personal” should be “personnel”.

Ok! Corrected.

Line 235. Naoplankton should be nanoplankton.  

Ok! Corrected.

Table 1. Check that information in certain columns is cutoff/incomplete visually within the information cell in the table. This was especially noticeable for information in the first column.

Ok!, we have remade this table.

Line 254. Donoflagellates should be dinoflagellates

Ok! Corrected.

Confusing mixing and matching of references to the same transects: Capre Gata vs. Cape Gata vs. CG.

 Ok. Capre Gata was just a spelling error. It has been corrected.

Figure 8, 9, 10, 11 –

First - why do the lines connect some data no matter how much time passed between them while not connecting other segments of the graphs. A more effective presentation would be to remove the lines connecting the data, then add in the trend line with confidence intervals.

Ok. We have followed the reviewer suggestion. We have removed the lines connecting the data points and we have simply included the lines representing the linear trends.

Second - Give complete figure descriptions to each figure. Imagine a reader might extract a single graphic to reference elsewhere. A figure description that says, for example in Fig 9 “Same as Fig 8 but for M2” says nothing about what information is actually being portrayed about plankton community group time series for Alboran Sea station (M2) and the trends on their residuals from a linear trend analysis, trend significance assessed at P<0.05. (Or, similar language. This is a just suggested form of text that should be associated with each figure.). Pay attention to this for all your graphics in the document. There is another solution to help this issue along per my third suggestion.

The reviewer is right. We have corrected this in all the figures. Now each figure has its own legend and it is not used "the same as in figure..." anymore.

Third – If you advance to graphics with data and trend lines, you should be able to shrink the size of each graph and potentially get all of them on 1 page. Alternatively, you could produce a single graph example in the main body of the text here and then provide a supplemental file of the remaining graphics.

We have merged figures 8 and 9 into a new figure 8, and figures 10 and 11 into the new figure 12. We could not reduce the size of the plots more than this.

Finally – if you have regression analyses and significance tests for all these data, a common summary table of the regression equation and significance test results (p-values) allows a more concise summary to the reader than noting slopes for each individual assessment in the text. Your text then can focus up one level from individual slopes to describe the range of slopes of change in abundance residuals for example and the average of the slopes, etc.

We have included a new table S5 in supplementary material with the linear trends and the confidence intervals at the 95 % confidence levels for all the depths at stations P2, M2, V2 and B1.

Throughout the document, I would suggest spelling out “small flagellates” rather than some places using the text small flagellates and other places using “small flagel” as some mysterious abbreviation of the full title of the group.

Ok! we have changed this abbreviation by the complete name: small flagellates in all the figures. We have also changed abbreviations for Prochlorococcus and Synechococcus by the full name in figures 2, 3, 4 and 5.

Line 393 18.3 not 18,3

Ok! Corrected.

Line 394 Update punctuation in the sentence a little here. One approach is to use brackets, such as “The remaining groups (doliolids,…,sychophysans) were less abundant…”.  Alternatively, you could get rid of the use of brackets altogether by eliminating the phrase “The remaining groups” and just state “The dolioids,…, and sychophysans were less abundant…”

 We have followed the reviewer's suggestion and we have used brackets.

Figure 12. I appreciate the single season focus of the graphics. The presentation is nice. However, I suggest the information of four seasonal graphics should be condensed into one here. I recommend using the same base picture of the sampling region, but instead of 4 seasonal pie charts on 4 separate figures, produce a stacked bar chart for each location with 4 bars representing each of the 4 seasons for a location, and the segments of each stack represent the % of the taxonomic groups counted within each season. Such a presentation should work well for you and obviously save you space.

 We have tried to follow the reviewer's suggestion but the result was not clear enough. So, we have preferred to maintain the current figure structure for the clarity of the plot.

I am unfamiliar with the abbreviation “cel./ml”.  I have always seen this spelled out as cells/ml (as you did in line 630) or cells*mil-1. I would say the same for your use of ind./ml. I defer to the editor for a final decision about which presentation of these units is most appropriate for this journal.

 We have always seen both forms. We would change it in the case that the editor considers that it should be changed.

Line 448. “Stormy winter activity…” – I don’t know what this means. Do you mean wind storms, sand storms, rain storms, snow storms? Or do you mean something like “Winter weather, with its associated long durations of high wind speeds and increased precipitation to the region compared to average spring, summer and autumn weather conditions,…”, or do you mean something else? Please be more descriptive and specific here. I think your text around line 540 gets at the type of translation for “stormy conditions” that would be helpful here. Thank you.

The reviewer is right. The main mechanism in this case is the wind storms, or simply the strong winds that can produce the mixing of the upper water column. In many areas of the Spanish Mediterranean, and also in many other regions of the Mediterranean, these winds are dry and cold because they blow from the continent. This produces a decrease of the surface temperature and an increase of the surface salinity that can enhance mixing because of density instabilities and consequently convection. We have tried to be more specific in the new redaction of this paragraph.

Line 454-456. This information would be more valuable to the article if you lead into it with a couple of sentences that define the conceptual model for nutrient loading and availability in the water column for the Mediterranean Sea. 

You state:

“In winter, maximum phytoplankton abundances can occur in the surface layer. As the spring advances, a seasonal thermocline is developed, inhibiting the nutrient supply to the photic layer and nutrients are depleted by phytoplankton uptake [30] which in turn is grazed by zooplankton.”

In other temperate ecosystems, spring is a time of increased rainfall. Elevated rainfall increases the nutrient delivery to the main body of a sea or estuary. Fresher water moves out over the saltier water feeding the phytoplankton in the photic zone. Your statements here suggest the Mediterranean Sea productivity is dominated by the deep water delivery of nutrients to the photic zone when there are well mixed conditions throughout the water column, and that nutrient loading from the land and air is insufficient to significantly influence plankton production during portions of the year where the water column has structure (i.e., thermocline at minimum).  I believe that some brief text then that gives the reader this sort of overview of the annual water quality dynamics over an annual cycle relating seasonal nutrient loading sources with nutrient availability, contrasting a minimum impact of land-and-air based nutrient sources and the dominance of ocean-based nutrient delivery, feeding plankton production of the sea as understood in this point in time will bridge the excellent ideas you are sharing with an understanding about how your ecosystem behaves as important context to support your argument. Just a little up front presentation of the system appears to me as the link many of your discussion elements can tie back to throughout your article. You may even consider providing such a conceptual diagram of the seasonal nutrient availability and loading sources in your introductory material as a firm basis for how you link your findings back to the understanding about the best understanding for how the ecosystem functions at this time.

Certainly this is a very interesting idea. In the case of RADMED area, only the river Ebro can supply nutrients to enhance the primary production of coastal waters. But there are other mechanisms that can drive the phyto and zooplankton dynamics. We accept that some information about these nutrient cycles would complete the present work. On the other hand the reviewer suggested to include a conceptual diagram. We have included all this new information in the supplementary material figure 1. 

Line 460-463. Is the seasonal succession of zooplankton species typical or unusual for inland seas? Can you please provide supporting references that give context to this statement as to whether this is a normal or abnormal successional progression with respect to zooplankton ecology? Otherwise, the statement is a result that belongs in the results section and not much of a discussion issue here.

Line 471. What does it mean to “relax oligotrophy”? Do you mean there is ” a hydrodynamic process that can enhance nutrient availability in the photic that results in elevated plankton production”?  

Yes. In fact, this is the meaning. These mechanisms enhance nutrient availability and increase the primary production, but we would not use the term "elevated production". In the references cited and in our opinion, this nutrient supply simply makes conditions less oligotrophic than in other areas of the Mediterranean where such mechanisms do not act. We have changed the redaction and hope that it is more clear now.

Line 490. Given the impacts of climate change on slowing certain ocean circulation patterns like the Gulf Stream in the Atlantic Ocean for example, I would caution you to describe any circulation feature as “permanent”. There may be limited evidence of change in these circulation features for the Mediterranean Sea at this time, however, climate change effects may impact the functioning of such processes in the future. (I see you actually reference climate change effects on water column behavior in line 612.) Perhaps you have worked with others that have produced model scenarios of increasing temperatures, the impact of more freshwater in the Atlantic from ice melt, projected changes in precipitation patterns to your region, etc. that might suggest there is no evidence circulation features will be impacted significantly by climate change in the near future (e.g., 50-100 year projections). That would be helpful to reference. In the meantime, consider using a different term than “permanent”.

We understand the point of view of the reviewer. And certainly we think that he/she is right: In the present climate change scenario, we cannot state that any circulation pattern is permanent, if we understand as permanent something that is not going to change in the next decades or centuries. It is true that the warming of surface waters and the decrease of the salinity in the northern Atlantic could alter winter convection and slow down the thermohaline circulation. The case of the Mediterranean Sea is not so clear because model projections show both an increase of the temperature and an increase of the salinity that could compensate the effect of temperature on density. Anyway, this is not our point. When saying "permanent" we simply referred to a process that does not depend on the seasonal cycle and that acts throughout the year. We have to accept that this was not clear in the manuscript and we have changed the redaction.

Line 551. A single sentence as a paragraph is rather unusual for an article. You have several in succession here. I believe what you want here is something restructured into a single paragraph, more like this:

The results from the statistical analysis presented in this work suggest that there are three main areas in the Spanish Mediterranean waters.  First, the most productive waters located in the western part of the Alboran Sea where primary production and the presence of large cells such as diatoms are influenced by winter mixing, but also by other fertilizing processes linked to the dynamics of the Strait of Gibraltar [52,61] and the cyclonic circulation areas [62,30]. The consequence is a high phytoplanktonic biomass throughout the year. These high biomass values are also located at the very surface waters, indicating the nutrient supply to surface or subsurface depths. Second, the poorest waters are those located to the south of the Balearic Islands and Cape Palos where small flagellates dominate during all the seasons. And third, the Catalan Sea, where the described oligotrophy could be partially “relaxed” by strong wind episodes, being winter mixing stronger than at the Balearic Sea.

We have to admit that this paragraph is improved with the redaction proposed by the reviewer. We have changed it and used the paragraph proposed.

Line 577. For this paragraph, I would recommend starting the paragraph emphasizing the complete long time series you were able to work with first, then refer to the abundance of limited data sets, and end the paragraph similar to what you have done with suggestions for better continuity throughout the region to enhance understanding of zooplankton dynamics.

 Ok!, we have started this paragraph with a sentence emphasizing the length of some of these time series, which in some cases are longer than 20 years.

Line 584, 585 – no need for identifying the abbreviations again at this point. That should be accomplished early in the article.

The reviewer is right. The correspondence between transects and their abbreviations had already been explained and there is no need for repeating it here. So, we have suppressed it.

Line 630. “Cuasi-“ should be “quasi-“

 Ok! Corrected.

Final sentence. Please rephrase this sentence. It is choppy in its presentation.

  We admit that we do not find a better way to express this sentence. We have maintain it because we are aware of the need of monitoring programs. It is important to emphasize it because of the many difficulties for maintain such programs.

Below – this document should be a useful reference to include in your article. Nutrient Cycling in the Mediterranean Sea: The Key to Understanding How the Unique Marine Ecosystem Functions and Responds to Anthropogenic Pressures. By Helen R. Powley, Philippe Van Cappellen and Michael D. Krom Submitted: December 2nd 2016 Reviewed: September 7th 2017 Published: November 8th 2017. DOI: 10.5772/intechopen.70878

Ok, we have found very intersting and well written this work and we have included it as a new reference. Thank you-

Reviewer 2 Report

According to the authors “The goal of the present work is to complete the previous ones and provide baseline data that could be used for regionalization of the pelagic ecosystem, initialization or validation of ecological models or simply as a reference for future works.” Indeed, the manuscript provides a large amount of data which are used for a description mostly of the phytoplankton groups spatial and temporal variability. A minor effort was payed to investigate statistically long-term variability, while no attempt was made to investigate spatial differences by statistical tests. Therefore, a purely descriptive manuscript could not be accepted for publication in an international journal.  In addition, there are several problems which reinforce the view of rejection. Namely:

1.      The first part of the abstract does not provide information for the obtained results, but constitutes a part of an introduction.

2.      In the methodology, a more detailed table with the performed cruises per transect is necessary. Few sentences in the discussion should be moved in the methodology

3.      The past tense should be used in the entire manuscript, instead of the present tense.

4.      The term Bacteria concerns both autotrophic and heterotrophic bacteria. When referring to Synechococcus and Prochlorococcus, you should use the term “cyanobacteria” or “picoprocaryotes”; use the same term in the entire manuscript.

5.      Results: The gradual decrease of diatoms abundance and relative abundance was not evident by the presented data, e.g. the abundance of diatoms was higher at BNA2 than at C20 and B2.

6.      In the discussion, there are many repetitions of the proper results, by mentioning also the relevant figures. The authors state there was a southwest‐northeast gradient in the trophic conditions, but this is not supported by results since no chla and nutrients data were presented, or by references. Moreover, the gradual southwest‐northeast decrease of diatoms is not supported by the presented results. The authors stated that there are differences within areas or between areas, but they do not clearly write which are exactly these differences and how do they comment them. They refer to statistical analysis for the distinction of three areas, but no such analysis was performed. There are a lot of statements which need to be supported by references. The seasonal cycle of zooplankton groups does not depend only on the available food, but on several parameters.

7.      Language needs a deep revision. Some sentences are unclear.

More detailed comments are given in the attached file (word version of the manuscript).

However, the available data sets are valuable and they could provide interesting and sound results. More efforts are necessary for the elaboration of the data, the statistical analyses of the data (e.g. ANOVA or Kryskal-Wallis test for detecting spatial differences), a more clear and concrete discussion and a well written manuscript.

Author Response

First we thank the reviewer for her/his comments. The reviewer's comments are in bold and the answers are below each comment.

Are the conclussions supported by the results? ( ) ( ) (x) ( )

Comments and suggestions for authors

According to the authors "The goal of the present work is to complete the previous ones and provide baseline data that could be used for regionalization on the pelagic ecosystem, initialization or validation of ecological models or simply as a reference for future works". Indeed, the manuscript provides a large amount of data which are used for a description mostly of the phytoplankton groups spatial and temporal variability. A minor effort was payed to investigate statistically long-term variability, while no attempt was made to investigate spatial differences by statistical tests. Therefore, a purely descriptive manuscript could not be accepted for publication in an international journal. In addition, there are several problems which reinforce the view of rejection. Namely:

There is no doubt that we have not been able to transmit the real objective of the present work. We hope that it is clearer in the new version.

First we would like to say that international journals have hundreds, if not thousands of descriptive works. We are sure that the reviewer is aware of it and probably she/he has read many of them, but if necessary we could provide a list of them just for the Western Mediterranean.

Anyway, we would like to emphasize that this is not simply a descriptive work. As stated in the abstract, the average abundances for the different phyto and zooplanktonic groups presented in this work, are based on long time series. As also explained in the introduction, many descriptive works attempt to do such a description using data from just 4 data sets or campaigns (or less). In such cases, a description of the abundances of phytoplanktonic or zooplanktonic groups, or nutrient concentrations, etc. are presented, but it cannot be stated that those abundances and concentrations are representative of the mean or average values (or the most likely values if the statistical distribution is normal). Neither can such works provide estimations of the uncertainties associated to the average or mean values, as no time series are available. For all these reasons, we consider that the presentation of these average values for each season is a very valuable information that merits publication.

Furthermore. Most of the descriptions of the abundances of phyto and zooplanktonic groups in the Mediterranean Sea that can be found in the literature, are based on two or four campaigns, and cover smaller geographical areas than the one covered in the RADMED monitoring and presented in this work. Some examples are numerous works devoted just to the Almeria-Oran front, or just to the northwestern Alboran Sea, or just to the Catalan continental slope, etc.

Another reason for presenting these average seasonal values, with their uncertainties, is that they allow us to estimate linear trends. It is true that the results presented here are incomplete, because some of the time series are not long enough, but they are the longest ones that do exist in the Spanish continental shelf and slope, which cover a large percentage of the western Mediterranean coastal waters. Please, take into consideration that, as explained in the introduction, there are very few monitoring programs in the Western Mediterranean. In our opinion this gives further support to the present work.

Finally, the reviewer thinks that the effort devoted to analyze spatial differences is not enough. In her/his final lines the reviewer says that an ANOVA or Kruskal-wallis test could be performed. We agree with the reviewer about the need for further analyses. But this will be the topic for a future work that hopefully will be published in an international journal as Water. The first step, after analyzing thousands of data was to provide the mean values, standard deviations and ranges of variability that characterize each area. The tables provided can be used for consultation for future works. This was more difficult to do when the available information came from just one data set per season and no uncertainties could be estimated. Second, a work dealing with a regionalization of the Spanish Mediterranean needs more than an ANOVA and that is the reason why we think that this merits a different work. Please, take into account that an ANOVA is simply the comparison of several sets of data in order to establish if the mean values of all the groups are the same or there is any of them that is different (comparing the variances within the groups and between the groups). If the data are not normally distributed this test is made using a non-parametric one as kruskall-wallis. Which data set do we use for these tests: Diatom abundances at the sea surface and the different  oceanographic stations? Or diatom abundances at 10 m depth? Or small flagellate abundances at 75 m? Or the copepod abundances? or the appendicularian ones?

The question is that we have so many time series that it is difficult to reduce the dimensionality of the statistical problem. A cluster analysis could be used, or a Principal Component Analysis, etc. We sincerely believe that the present work is long and dense enough and all these question will be addressed in the near future in a separate work.

1.The first part of the abstract does not provide information for the results, but constitutes a part of an introduction.

We think that the first part prepares the reader for the second one where the  new results and objectives of the work are presented. We could remove this part if the editor also considers it, but the abstract is not very long and we think that it could remain as it is.

2. In the methodology, a more detailed table with the performed cruises per transect is necessary. Few sentences in the discussion should be moved in the methodology.

We agree on this point and a new table 1 has been prepared for the new version.

3. The past tense should be used in the entire manuscript, instead of the present tense.

We agree again and the past tense has been used in the methods and results sections.

4. The term bacteria concerns both autotrophic and heterotrophic bacteria. When referring to Synechococcus and Prochlorococcus, you should use the term "cyanobacteria" or "picoprocaryotes"; use the same term in the entire manuscript.

We also agree. We have used the term prokaryote pico-phytoplankton, except when a different analysis is carried out for Prochlorococcus and Synechococcus. In that case the name of these two groups are used.

5. Results: The gradual decrease of diatoms abundance and relative abundance was not evident in the present data, e.g. the abundance of diatoms was higher in BNA2 than at C20 and B2.

Ok! We have made a deep revision of the manuscript and hope that it is clearer now. There is a southwest-northeast gradient as the abundances decrease along the Alboran Sea toward the east, and then, the abundances to the north of Cape Gata are always lower than those of the Alboran Sea. This is the sense of this sentence. then, it is true that, during some seasons (not along the whole year), there is again a slight increase of abundances in BNA transect, but never reach the Alboran Sea values. We hope that this is better explained in the new version. In fact we explain that there are three different areas and we explain that the area around Barcelona and Tarragona is similar to the Balearic Islands but with slight differences.

6. In the discussion there are many repetitions of the proper results, by mentioning also the relevant figures. The authors state there was a southwest-northeast gradient in the trophic conditions, but this is not supported by results since no chla and nutrients data were presented, or by references within areas or between areas, but they do not clearly write which are exactly these differences and how do they comment them. They refer to statistical analysis for the distinction of three areas, but no such analysis was performed. There are a lot of statements which need to be supported by references. The seasonal cycle of zooplankton groups does not depend only on available food, but on several parameters.

Ok!Once again the reviewer is right. We have included the reference García-Martínez et al. (2018) when this statement is done.

We have already explained that the statistical analyses are the estimation of the mean values, standard deviations and ranges of variability (maximum and minimum values recorded) for each station along the different areas covered by RADMED monitoring program.

It is true that the zooplankton cycle depends on many variables. We just propose an hypothesis based on the data at hand and some previous works.

7. Language needs a deep revision. Some sentences are unclear.

Ok! A bilingual colleague has reviewed the manuscript. We have also followed the suggestions and corrections made by both reviewers.

More detailed comments are given in the attached file (word version of the manuscript).

However the available data sets are valuable and they could provide interesting and sound results. More efforts are necessary for the elaboration of the data, the statistical analyses of the data (e.g. ANOVA or Kryskal-Wallis test for detecting spatial differences), a more clear and concrete discussion and well written manuscript.

The reviewer is right. This data set can provide and will provide much more information after new statistical analyses. Not just an ANOVA, more sophisticated analysis methods are needed. This will be the subject of a future work. Nevertheless we believe that the present analysis was necessary as a first step.

The reviewer has submitted a revised version with numerous corrections and suggestions. We have followed most of them and corrected within the manuscript, therefore is difficult to answer here to these comments. Nevertheless, find below some of these comments:

Comments and corrections highlighted in the document water-422368-review-1.docx.

-We have included a reference for the solubility pump.

-We have included references for the temperature, salinity, phyto and zooplankton seasonal cycles in the introduction.

-We have deleted some references that were cited in a wrong style, e.g. García-Martínez et al., 2018.

-We have included information related to all the CTD models.

-We have changed stations 2 and 4 by the second and fourth stations for each transect as suggested by the reviewer.

-The reviewer asked for a clarification about the sentence "average seasonal cycle...", we have explained this better.

-The reviewer suggested that when dealing with cyanobacteria, always the same term should be used, instead of autotrophic bacteria, cyanobacteria, prokaryote picoplankton, etc. We have followed this suggestion and we have tried to use the term: prokaryote pico-phytoplankton when this term was referred to both Prochlorococcus and Synechococcus. When a distinctive analysis was carried out for Prochlorococcus and Synechococcus, these names are used.

-Table 1 has been re-made.

-The reviewer suggests that the sentence eastward gradient should be clarified, indicating "from which transect to which transect". This has been done.

-Reviwer suggested to change abundances by relative abundances. We have done it.

-Concerning the sentence: "most productive waters of the Alboran Sea and those more oligotrophic ones in the Balearic Sea", the reviewer states that this needs some references as this can only be established using nutrient and chlorophyll data. Ok! we have provided this reference in this sentence:[34] García-Martínez et al., 2018. In this work, nutrient and chlorophyll data are analyzed in the same region.

-The terms late winter and early spring should not be used as the sampling is not monthly. The reviewer is right and this has been corrected using just winter and spring.

- The reviewer suggested to move the sentence: "abundance of diatoms reach 300 or 400 cel/ml" from discussion to results. Ok! we have done it.

-The reviewer has suggested to change "large cells" by diatoms as some dinoflagellates can also be large. This has been corrected.

-We have used the past tense for the methods and results section as suggested by the reviewer in his/her revised manuscript.

-The reviewer also suggested that the reference Thiebault et al. (1994) when dealing with the summer zooplanktonic abundance increase is not appropriate. He/she is right and we have changed it by Yebra et al., 2017.

-The reviewer points out that the term "vertical distributions" is used for zooplankton data. This is not correct, and has been corrected eliminating the word "vertical" in this case.

-The reviewer calls our attention on the sentence: The intensity of winter winds and convection

 decreases southward from the Gulf of Lions towards the Catalan Sea[r1] . This seems to be the cause for the higher winter microphytoplankton abundances at the Barcelona transect where diatoms become the main group and also the importance of this group in Tarragona transect[r2] . As pointed out by [26] nanopankton seems to be less variable. Thus, the main differences between the different geographical areas and the larger temporal variability are associated to the microplankton size fraction[r3] 

This was not the meaning of this sentence. We do not state that the winter convective processes in the Gulf of Lions are the responsible for the higher productivity of waters in Catalonia waters when compared to those at the Balearic Islands. We simply state that the intensity of winds and the convective processes are vetry intense in the Gulf of Lions, where deep water can be formed, and then decreases southwards. In Catalonia the intensity is lower, and in the Balearic Islands is much lower, with the possible exception of Menorca. In fact, in Catalonia waters there can be intermediate convection, being this process very exceptional to the south of the Ibiza Channel.

- There are many other suggestions and corrections in the word document attached by reviewer #2. Most of them have been corrected directly in the new version.

Add reference. 

The deep convection in the Gulf of Lions occurs beyond the continental slope, whereas the transects T and BNA are positioned over the continental shelf. It is not clear how the injected nutrients in the photic layer of the G.of Lions could reach the areas of above transects

Not clear. Do you mean that nanoplankton was less variable than microplankton between areas?

Round 2

Reviewer 1 Report

Thank you for your excellent work and for considering the recommendations provided during the review process. Your detailed reply to the suggestions was excellent, comprehensive and very much appreciated. With all your work the quality of the resulting manuscript is substantially improved from what was already a very good manuscript. Congratulations.

Two final points.

Line 495. I think you accidentally deleted 'e' from 'the' at the end of that line.

Supplemental graphics

First sentence in SA1 figure title. I think you want to say "enhances" instead of "enhance."

Graphics. I like these in concept. I grasp them at about 90% of what I think I see there. I am kindly recommending minor changes in their presentation to make them less cluttered and focus on the dynamic information rather than a mix of static and dynamic. 

I see a blue line in space - I don't understand why that is there. Does it need to be in these charts? My understanding doesn't change if I mentally remove them so I don't understand why they are there. Maybe they can be deleted.

I see a subset of information in each graphic that is static. This is cluttering the important dynamic information you seem to want to get across in your presentation. Ideally, I think if you could just include those static numbers in a legend or in the text of the figure title, it will make the rest of figures cleaner and more understandable. Specifically, every chart has 2 blue dotted reference lines of 4 ml/l and 5 ml/l, a green dotted reference line of 1 mg/m3 and a brown dotted line of 1 microM. Because this information is static and common to all the charts, if you describe it once in the figure information (or a legend) and remove those numbers and arrows from each chart, you will make the presentation more concise and cleaner to read.

If you do that sort of adjustment with the figures, now we can focus on the dynamic information between the locations, the integrated water quality measure comparisons. And without all the static information in the figures, you could move the nitrogen and phosphorus numbers up under the chla number so all the text information is together above the chart information rather than split apart around the chart.

That is one way I see the supplemental presentation could be made more clear and should only require a small amount of time to improve the graphics.

Good luck! Once again, excellent work, I enjoyed this review covering the RADMED program and seeing the relatively uncommon long term aquatic living resource time series brought to life in your analyses and reporting. Just a few minutes work on the supplemental graphics for me seems to complete the work needed for publication now if the editor agrees. Take good care and keep up the excellent monitoring and analysis work on the Mediterranean Sea!  

Author Response

Once again we would like to thank the reviewer very sincerely for his/her constructive comments.

We have corrected the spelling mistakes that the reviewer commented and some other mistakes that we have found.

We have also re-made the figure S1 with the conceptual scheme for the RADMED area. As suggested by the reviewer we have made it clearer suppressing some of the legends with static information. As suggested by the reviewer this information is now provided in the legend of the figure. All the figures providing dynamical information are included together in a single legend inserted in the plots for each area, as suggested by the reviewer.

The reviewer also asked for the meaning of the blue horizontal bar. This represented an schematic of the horizontal axis for the dissolved oxygen concentration. It was not clear in the previous version of this figure because it was displaced upwards from the vertical axis. Now this has been corrected and we think that its meaning is clear.

We sincerely appreciate the comments made by the reviewer.